# $\mathrm{SE}(3)$-Equivariant Diffusion Policy in Spherical Fourier Space

**Xupeng Zhu** [1] [2]   **Fan Wang** [3]   **Robin Walters** [2]   **Jane Shi** [3]

## Abstract

Diffusion Policies are effective at learning closed-loop manipulation policies from human demonstrations but generalize poorly to novel arrangements of objects in 3D space, hurting real-world performance. To address this issue, we propose Spherical Diffusion Policy (SDP), an $\mathrm{SE}(3)$ equivariant diffusion policy that adapts trajectories according to 3D transformations of the scene. Such equivariance is achieved by embedding the states, actions, and the denoising process in spherical Fourier space. Additionally, we employ novel spherical FiLM layers to condition the action denoising process equivariantly on the scene embeddings. Lastly, we propose a spherical denoising temporal U-net that achieves spatiotemporal equivariance with computational efficiency. In the end, SDP is end-to-end $\mathrm{SE}(3)$ equivariant, allowing robust generalization across transformed 3D scenes. SDP demonstrates a large performance improvement over strong baselines in 20 simulation tasks and 5 physical robot tasks including single-arm and bi-manual embodiments. Code is available at https://github.com/amazon-science/Spherical_Diffusion_Policy.

## 1. Introduction

Diffusion Policy (Chi et al., 2023) has emerged as an effective method for learning closed-loop policies from human demonstration. This success is based on the ability of Diffusion models (Ho et al., 2020) to approximate multi-modal human demonstrations (Mandlekar et al., 2021). A particularly challenging aspect of real-world robotic manipulation, which is often underrepresented in synthetic benchmarks, is that objects may be found in a wide range of 3D poses.

[1]Work was done as an intern at Amazon Robotics [2]Khoury College of Computer Science, Boston, Massachusetts, USA [3]Amazon Robotics, Boston, Massachusetts, USA. Correspondence to: Xupeng Zhu <zhu.xup@northeastern.edu>.

*Proceedings of the $42^{nd}$ International Conference on Machine Learning*, Vancouver, Canada. PMLR 267, 2025. Copyright 2025 by the author(s).

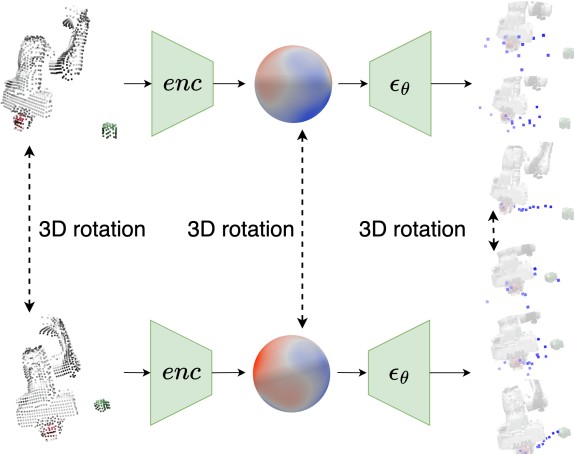

*Figure 1.* SDP enforces that the policy is $\mathrm{SO}(3)$ equivariant. Specifically, in the second row, an $\mathrm{SO}(3)$ rotation that is applied to the scene leads to an equivalent rotation on the latent spherical Fourier features in the neural networks $enc, \epsilon_\theta$, and on the generated trajectory (blue dots). Fourier features are visualized as spherical signal.

Consider, for example, grasping a dish that is randomly placed in the sink, threading a nut onto a bolt with random orientation, or wiping the curved surface of a car. Diffusion Policy may struggle to attain robust 3D generalization without training on a large amount of costly human demonstrations to exhaust the possible 3D arrangements of the scene.

We propose Spherical Diffusion Policy (SDP), a Fourier space $\mathrm{SE}(3)$ equivariant method that automatically adapts to changes in the scene. SDP improves on recent works in equivariant diffusion policy learning which are limited to $\mathrm{SO}(2)$-equivariance (Wang et al., 2024b), equivariant to only single-object transformations (Yang et al., 2024a; Tie et al., 2024), or computationally heavy (Tie et al., 2024). In contrast, our method is light and $\mathrm{SE}(3)$ equivariant across multiple objects, allowing it to perform more complicated tasks with less engineering. SDP achieves translational invariance by formulating states and actions in the gripper frame (Chi et al., 2024). Figure 1 illustrates the $\mathrm{SO}(3)$ equivariance of the proposed method. If the scene is transformed by a 3D rotation, then the denoised action trajectory

will be rotated by the same rotation. Since this equivariance is embedded in the neural network, it does not rely on additional data to train and thus achieves high sample efficiency. The equivariance constraints lead to provable SE(3) generalization to transformed scenes.

The contributions of this work are:

1. a novel method, Spherical Diffusion Policy, which is equivariant to 3D rotations and invariant to 3D translations enabling generalization to unseen scenes,

2. a novel spherical FiLM layer for SO(3) equivariant conditioning,

3. a novel spherical denoising temporal U-net for denoising trajectories with spatiotemporal-equivariance,

4. theoretical validation that SDP is equivariant,

5. empirical validation of SDP through extensive experiments that include 20 simulation and 5 physical tasks including single-arm and bi-manual embodiments.

## 2. Background

**Diffusion Policy** is a closed-loop imitation learning method that learns a policy $\pi(s) = a$ that maps states to action trajectories from expert demonstrations. The states $s$ consist of camera observation $o$, e.g. images or voxels or point clouds, and the end-effector's 6DoF pose (3D translation and 3D rotation) $e_T, e_R$ and aperture $e_{grip}$. The action $a$ specifies the 6DoF pose $a_T, a_R$ and gripper aperture $a_{grip}$. The policy takes an input a history of $h$ states $S_t = [s_t, s_{t-1}, \ldots, s_{t-h+1}]$. The output is an action sequence of $r$ actions $A_t = [a_t, a_{t+1}, \ldots, a_{t+r-1}]$.

Diffusion Policy (Chi et al., 2023) leverages Diffusion Models (Ho et al., 2020; Song et al., 2021) to learn from multimodal human demonstrations (Mandlekar et al., 2021). Diffusion policy infers actions by sampling $A_t^K$ from a uniform Gaussian noise, then performing $K$ iterations of denoising, producing $A_t^K, A_t^{K-1}, .., A_t^0$. The final iterate $A_t^0$ is the output action. The denoising process is defined by:

$$A_t^{k-1} = \alpha\big(A_t^k - \gamma\epsilon_\theta(S_t, A_t^k, k) + z\big), z \sim \mathcal{N}(0, \sigma^2 I) \quad (1)$$

where $\epsilon_\theta(S_t, A_t^k, k)$ is a learnable denoising function, parameterized by $\theta$, that estimates the noise $\epsilon^k$ based on the state $S_t$, the noisy action $A_t^k$, and the step $k$. The parameters $\alpha, \gamma, \sigma$ define the noise schedule and functions of the denoising step $k$. Finally, the denoising function is trained to predict the noise added to the expert action:

$$\mathcal{L} = \big\|\epsilon_\theta(S_t, A_t^0 + \epsilon, k) - \epsilon\big\|^2. \quad (2)$$

**Equivariance** describes the property of a function which commutes with the transformations of a symmetry group

$G$: $\rho_{\text{out}}(g)f(x) = f\big(\rho_{\text{in}}(g)x\big)$, for all $g \in G$. Here, the $\rho$s denote group representations, mapping each group element to an invertible matrix (Serre et al., 1977). The 2D special orthogonal group SO(2) describes planar rotations and its subgroup $C_n$ discretizes SO(2) into $n$ rotations. Similarly, SO(3) describes 3D rotations. We denote the group of 3D translations $\mathbb{T}(3)$. The Special Euclidian group SE(3) = SO(3) $\ltimes$ $\mathbb{T}(3)$ includes both 3D rotations and translations. For any group, the trivial representation $\rho_0$ assigns the identity matrix $\rho_0(g) = I$ to each group element. This makes invariance a special case of equivariance where the output representation $\rho_{\text{out}} = \rho_0$. For SO(3), there are higher-dimensional representations $\rho_1, \rho_2, \ldots$ that will be introduced later. Representations can be combined by direct sum $\rho(g) = \rho'(g) \oplus \rho''(g)$, where $\rho'(g)$ and $\rho''(g)$ are diagonal blocks in $\rho(g)$.

**Equivariant Policy Learning** assumes the policy is equivariant $\pi(gS) = ga, g \in G$, where $G$ could be SO(2) group or SE(3) group. One way to achieve equivariance is by recognizing and modeling an equivariant function using equivariant neural networks. (Ryu et al., 2024) states that for Brownian Diffusion on the SE(3) manifold, if the target function $\pi(S) = a$ is equivariant, then the denoising function $\epsilon_\theta$ is also equivariant: $\epsilon_\theta(gS, gA, k) = g\epsilon_\theta(S, A, k)$. EquiDiff (Wang et al., 2024b) extends this open-loop equivariance (Ryu et al., 2024) into closed-loop setups, but it is limited to SO(2) equivariance. For an additional introduction, see Appendix C.1.

Another way to achieve equivariance is by canonicalizing the input $S$ and output $a$ of a neural network (Zeng et al., 2022; Wang et al., 2021; Jia et al., 2023; Chi et al., 2024). For example, if $a$ is a 3D translation, then canonicalizing $S$ involves translating it inversely so that the action is at the origin: $S^{\text{can}} = S - a, a^{\text{can}} = a - a, a \in \mathbb{T}(3)$. Intuitively, canonicalization eliminates the transformation applied to the state and action by always evaluating the state in the canonicalized view. Refer to Appendix C.2 for proof.

**Spherical Harmonics (SH)** are functions on the sphere $Y_l^m: S^2 \rightarrow \mathbb{R}$ which give an orthonormal basis for the function space $L^2(S^2, \mathbb{R})$. They are indexed by degree $l \in \mathbb{Z}_{\geq 0}$ and order $-l \leqslant m \leqslant l, m \in \mathbb{Z}$. A spherical function in spatial space can be transformed into the frequency domain by a spherical Fourier transform: $\mathcal{F}: f \mapsto \{c_l^m\}$, where $c_l^m$ are Fourier coefficients. Inversely, the inverse spherical Fourier transform $\mathcal{F}^{-1}$ converts the Fourier coefficients to the spatial value: $f(u) = \sum_{l=0}^{\infty} \sum_m c_l^m Y_l^m(u)$. Spherical functions and SH are SO(3) steerable and thus suitable for SO(3)-equivariant networks. Essentially, a rotation of $f$ in spatial space is equivalent to a rotation of $c_l^m$ in the frequency domain by the Wigner D-matrices $D_{mn}^l$, which is orthogonal. That is, $f' = g \cdot f, g \in$ SO(3) is equivalent to $c_l^{n\prime} = \sum_m D_{mn}^l(g)c_l^m$, where $c_l^{n\prime}$ are Fourier coefficients

of $f'$. For example, degree 0 ($\rho_0$) Fourier coefficients $c_0 \in \mathbb{R}$ are scalars that are invariant to rotation, and degree 1 ($\rho_1$) Fourier coefficients $c_1 \in \mathbb{R}^3$ are 3D vectors with Wigner D-matrix given by a standard 3D rotation matrix. A Spherical Fourier signal up to degree $L$ has $(L + 1)^2$ coefficients (Cohen et al., 2018; Bonev et al., 2023). SDP leverages this compact representation. Convolving two sets of Spherical Fourier signals (Cohen et al., 2018; Klee et al., 2023) leads to a signal over SO(3), which has $\sum_l^L (2l+1)^2$ coefficients, as adopted in ET-SEED (Tie et al., 2024).

**Equiformer (Liao & Smidt, 2023)** and EquiformerV2 (Liao et al., 2024) are SE(3) equivariant graph neural networks (GNN) (Passaro & Zitnick, 2023). In contrast to conventional GNNs that treat each node in the graph as a scalar, Equiformer attaches spherical features to each node. These features are compactly approximated by truncated Fourier coefficients, up to degree $l \leqslant L$. Messages are aggregated from neighbor nodes in the graph through the edges by equivariant graph attention. This is followed by an equivariant spherical linear and activation layer. The spherical linear layer treats degree $l$ Fourier coefficients as high dimensional vectors to perform a linear mapping in each degree separately. The spherical activation layer (Geiger & Smidt, 2022) performs inverse Fourier transform, then performs conventional activation point-wise on the sphere, and lastly converts the activation back to Fourier coefficients.

## 3. Related Works

**Closed-loop Robot Policy Imitation Learning** learns robot skills from human demonstrations through machine learning. Though it is a straightforward and general framework, facing multiple challenges. One challenge is the error compounding effect where the action prediction error causes future states to diverge from the training states and further exacerbate the next action prediction (Ke et al., 2021). To combat this, action chunking (Lai et al., 2022; Mandlekar et al., 2021; Chi et al., 2023; Zhao et al., 2023b) proposes predicting and executing a trajectory of actions instead of one step of action. Another challenge is to learn from multi-modal human demonstrations. Multiple methods are proposed to fit a multi-modal policy, including Gaussian Mixture Model (Mandlekar et al., 2021; Zhu et al., 2022b), Variational Auto Encoder (Zhao et al., 2023b; Mousavian et al., 2019), Energy-Based Models (Implicit Models) (Florence et al., 2022), and Diffusion Models (Janner et al., 2022; Pearce et al., 2023; Chi et al., 2023). Based on (Chi et al., 2023), this work leverages additional inductive bias – equivariance, to achieve significantly better performance.

**Equivariance on Robot Learning** Robotic policies operate in the 3D world, sharing rich symmetries. (Zhu et al., 2022a; Huang et al., 2022; Zhu et al., 2023; Hu et al., 2024) investigated equivariance in the grasp learning. (Wang et al., 2021; Huang et al., 2024c; Simeonov et al., 2022; Zhao et al., 2023a; Ryu et al., 2024; Huang et al., 2024a; Gao et al., 2024; Zhu et al., 2025b) developed equivariant open-loop policies. (Van der Pol et al., 2020; Wang et al., 2022b;a; Jia et al., 2023; Liu et al., 2023; Wang et al., 2024c;b; Yang et al., 2024b;a) verified effectiveness of equivariance in closed-loop agent. Among these works, (Zhu et al., 2022a; Zhao et al., 2023a; Jia et al., 2023; Liu et al., 2023; Huang et al., 2024c; Wang et al., 2024b; Zhu et al., 2025a) utilize discrete equivariance that suffers from discretization error. On the other hand, (Ryu et al., 2024; Huang et al., 2024a; Gao et al., 2024; Hu et al., 2024; Zhu et al., 2025b) leverages continuous equivariance but is limited to open-loop settings. Moreover, EquiBot (Yang et al., 2024a) are limited to degree $l = 1$ representation that suppresses rich information, and ET-SEED(Tie et al., 2024) uses heavy SO(3) irreducible representation that needs two-stage inference to alleviate computation burden. Furthermore, (Yang et al., 2024a;b; Tie et al., 2024) requires a segmentation pipeline engineered for each task to exclude everything but one object in the workspace. In contrast, our work is the first to leverage continuous and compact spherical Fourier features to achieve a SE(3) equivariant, end-to-end, and computationally efficient closed-loop policy.

**Diffusion Models and Equivariant Diffusion Models** Diffusion Models (Sohl-Dickstein et al., 2015; Ho et al., 2020; Song et al., 2021) are probabilistic generative models that demonstrated a strong capability modeling multi-modal distribution. Such capability is achieved by iteratively removing noise from an initial sample randomly drawn from an underlying distribution. Equivariance is introduced to diffusion models in (Xu et al., 2022; Hoogeboom et al., 2022; Yim et al., 2023) in the context of molecule generation. Diffusion Models are applied to robotics in open-loop settings (Ke et al., 2024; Ryu et al., 2024; Jiang et al., 2023; Urain et al., 2023; Huang et al., 2024b) and closed-loop settings (Janner et al., 2022; Pearce et al., 2023; Chi et al., 2023; 2024; Ze et al., 2024; Wang et al., 2024a; Liu et al., 2024; Brehmer et al., 2024). The most relevant works on equivariant Diffusion Policy include (Wang et al., 2024b; Zhao et al., 2025; Yang et al., 2024a; Tie et al., 2024), where (Wang et al., 2021; Zhao et al., 2025; Hu et al., 2025) is limited to discretized SO(2) equivariance. (Yang et al., 2024a; Tie et al., 2024) is SO(3) equivariant thus requiring engineering effort to segment everything but the target object, even though, these methods are designed to handle a single object in the scene. Moreover, (Tie et al., 2024) is based on heavy SO(3) irreducible representations and needs 2 stage diffusion process. In contrast, our method is SE(3) equivariant, based on compact yet expressive spherical representations that can end-to-end learning without task-specific engineering effort and generalize to multi-object tasks.

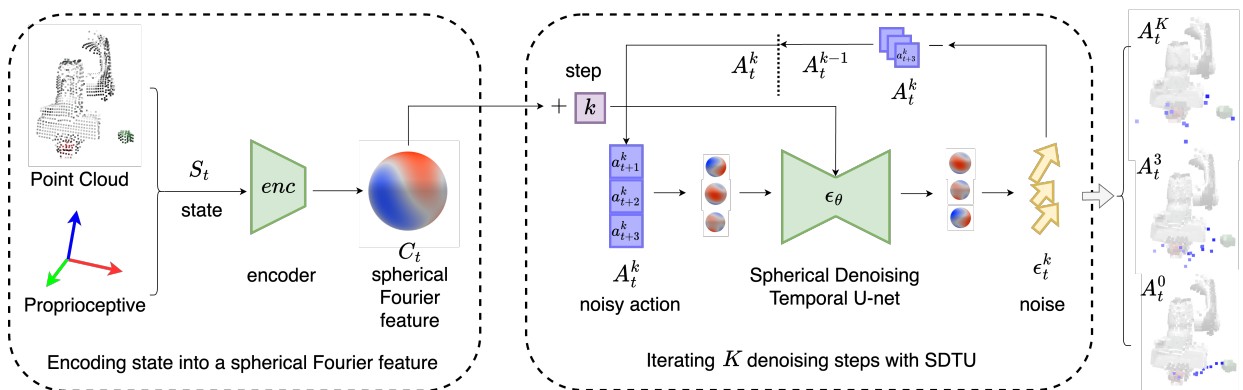

*Figure 2.* Method overview. During inference, SDP first embeds state $S_t$ into a spherical scene feature $C_t$ by the encoder $enc$. Then, SDTU $\epsilon_\theta$ estimates the noise $\epsilon$ based on the noisy actions $A_t^k$, step $k$, and the scene feature $C_t$. Later, the noise is subtracted from the noisy actions, generating cleaner actions $A_t^{k-1}$. This denoising process is performed for $K$ iterations, generating a clean trajectory $A_t^0$.

## 4. Method

### 4.1. Method Overview

The Spherical Diffusion Policy model maps observations to actions $\pi(S) = A$. We assume the optimal policy is SE(3) equivariant and enforce this assumption in the model. Specifically, we enforce rotation equivariance $\pi(gS) = gA, g \in$ SO(3), and translation invariance $\pi(tS) = A, t \in \mathbb{T}(3)$. The model is thus SE(3)-equivariant where $\mathbb{T}(3)$ acts trivially on the actions.

The rotational equivariance of $\pi$ is enforced by an equivariant denoising function $\epsilon_\theta$, as proven in (Ryu et al., 2024; Wang et al., 2024b). Specifically, we use an equivariant conditional denoising function $\epsilon_\theta(S, A + \epsilon^k, k)$ to estimate the noise for a noisy action $A + \epsilon^k$, the step $k$, and state $S$. We model $\epsilon_\theta$ using three components as shown in Figure 2: i) the spherical encoder embeds the state into a multichannel spherical scene feature $enc(S) = C$, and then ii) a spherical denoising temporal Unet (SDTU) estimates the noise from the noisy action and step, conditioned on the scene feature $\epsilon_\theta(C, A^k + \epsilon^k, k)$ using iii) spherical FiLM (SFiLM) layers to achieve this equivariant conditioning. Since these three components are equivariant, the denoising function is equivariant by composition.

Translational invariance of $\pi$ is achieved using a relative action formulation (Chi et al., 2024), which canonicalizes the state-action (Zeng et al., 2022) with respect to translations by centering the observation on the gripper and defining action positions relative to the gripper:

$$S^{can,i} = (O - e_T^i, e_T^i - e_T^i, e_R^i, e_{grip}^i)$$
$$A^{can,i} = (A_T^i - e_T^i, A_R^i, A_{grip}^i).$$

See Appendix C.2 for proof. For the single-arm setting, $i \in \{0\}$ denotes the gripper. Additionally, we propose bi-manual relative action representation. In this case, $i \in$ $\{0, 1\}$ and we canonicalize the state and action to the left $i = 0$ and the right $i = 1$ gripper's position.

### 4.2. Representing State and Action by Spherical Signal

In this section, we propose a spherical representation of the state and action for the policy. There are several advantages of using spherical Fourier features as latent features. First, the truncated spherical Fourier coefficients provide a compact approximation of spherical features and are compatible with SO(3) rotations, rather than computationally heavy SO(3) irreps used in ET-SEED (Tie et al., 2024). Furthermore, higher degree coefficients can represent finer details than EquiBot(Yang et al., 2024a) which adopted Vector Neuron (Deng et al., 2021) that only supports up to 3D vectors (analogous to type-$l = 1$), suppressing rich higher degree information in the latent features. For example, vector representations cannot capture spherical distributions with two distinct modes. Lastly, spherical features support equivariance to continuous group SO(3) (continuous rotation), in contrast to discretized group $C_8$ (discretized rotation) in EquiDiff (Wang et al., 2024b) which suffers from discretization error.

The end-effector state $e$, the action $a_t$, and the noise $\epsilon$ have the same geometric structure consisting of a 3D position, 3D rotation, and 1D gripper aperture information. We decompose the end-effector data as a 3D position vector, a $3 \times 3$ rotation matrix, and a 1D scalar. The rotation matrix can be viewed as 3 column vectors. We represent the position vector by a degree 1 vector, the rotation matrix by 3 degree 1 vectors, and the aperture as a scalar in the trivial representation. That is, $e, a_t, \epsilon \in \rho_{ee} = \rho_1^4 \oplus \rho_0$. Intuitively, the position and rotation matrix are rotated by the rotation matrix corresponding to the rotation of the state, while the gripper aperture stays unchanged.

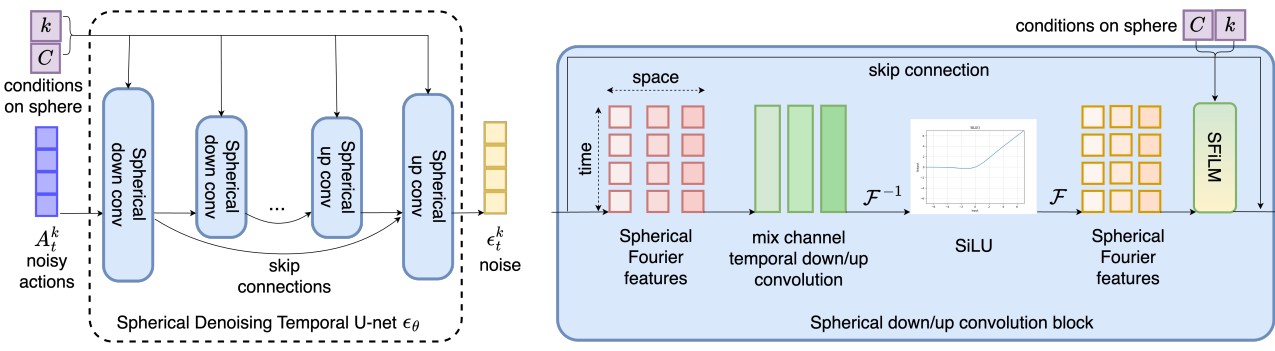

*Figure 3.* Spherical denoising temporal U-net (SDTU). Left: The SDTU $\epsilon_\theta$ estimates the noise $\epsilon$, based on the noisy actions $A_t^k$, denoising step index $k$, and the encoded scene $C$. The SDTU has a U-net architecture, with 4 spherical down or up convolution blocks. Right: details of a spherical down or up convolution block.

We adopt point clouds as an observation $o$ and treat color information as degree 0 spherical coefficients (same as (Ryu et al., 2024)), as the color is invariant to the point cloud rotation. The point cloud is encoded to a latent vector by a 5-layer ResNet (He et al., 2016) encoder $enc(\cdot)$. The encoder is implemented by EquiformerV2 (Liao et al., 2024) which extracts a high-degree spherical signal from the point cloud, see Appendix E for details. The robot state $e$ is concatenated to the output of the encoder yielding $C$.

### 4.3. Spherical Denoising Temporal U-net

The spherical denoising temporal U-net $\epsilon_\theta$ infers the noise from the noisy action $A^k + \epsilon^k$, denoising step index $k$, and state embedding $C$ as $\epsilon_\theta(C, A^k + \epsilon^k, k)$. The vector $C$ encodes the state in spherical Fourier space up to degree $L$. The input $A^k + \epsilon^k$ and the output are spherical signals in $\rho_{ee}$, as introduced in Section 4.2. The denoising step index is encoded using sinusoidal embeddings, treated as degree 0 features.

The SDTU is a 1D U-net with spherical Fourier features that are spatiotemporal equivariant. The temporal equivariance is achieved using 1D convolution along the time dimension $t$, as proposed in Diffuser (Janner et al., 2022). We incorporate an additional spherical Fourier dimension in the latent features to achieve spatial equivariance (SO(3) equivariance). This equivariance is enforced by mixing channel temporal convolution on each degree $l$ of the spherical Fourier coefficient:

$$h_{l,m,t}^o = \sum_{j=0}^{r} \sum_{i \in in} h_{l,m,j}^i w_{l,j-t}^{i,o}, \quad (3)$$

where $i, o$ indexing the input and the output channels respectively, $l, m$ is the degree and order of the spherical Fourier coefficient $h$, $in$ denotes all input channels. Subscript $j$ indexes the time in the prediction trajectories.

**Proposition 4.1.** *The mixing channel temporal convolution*

*in Equation. 3 is* SO(3) *equivariant:*

$$D_{mn}^l(g) h_{l,m,t}^o = \sum_{j \in T} \sum_{i \in in} D_{mn}^l(g) h_{l,m,j}^i w_{l,j-t}^{i,o}. \quad (4)$$

For proof see Appendix A.1, the proof essentially follows Schur's lemma (Schur, 1905), which states that any linear combination of SO(3) Fourier features is equivariant. This convolution is followed by spherical activation (Cohen et al., 2018; Geiger & Smidt, 2022) for expressiveness. Stride and transposed convolution are used for down- and up- sampling in the U-net, as in Diffuser (Janner et al., 2022). Spherical FiLM layers are adopted, allowing for equivariant conditioning, and are described in the next section. Figure 3 summarizes the SDTU.

### 4.4. Spherical FiLM Conditioning Layer

We propose equivariant spherical FiLM (SFiLM) layers to extend the Feature-wise Linear Modulation (FiLM) layer (Perez et al., 2018) used by Diffuser (Janner et al., 2022) into the spherical Fourier domain. The condition on sphere $C$ is projected into a scaling condition $\gamma$ and an offset condition $\beta$ by equivariant linear layers (Geiger & Smidt, 2022): $\gamma = \Gamma(C), \beta = B(C)$. Then, SFiLM conditions each degree $l$ separately. Specifically, SFiLM treats $\gamma_l, \beta_l$ as $2l + 1$ dimensional vectors, to modulate the hidden feature $h_l$, by projecting $\gamma_l$ onto $h_l$ as a scaling condition and adding $\beta_l$ as an offset condition:

$$\text{SFiLM}(h_l | \gamma_l, \beta_l) = \gamma_l^{\text{T}} h_l \frac{h_l}{||h_l||} + \beta_l. \quad (5)$$

SFiLM supports high degree Fourier coefficients for expressiveness, which differs from EquiBot (Yang et al., 2024a) that only supports degree 1 which drops rich information in the latent features.

**Proposition 4.2.** *The SFiLM layer in Equation. 5 is* SO(3)

*equivariant:*

$$D(g) \cdot \textit{SFiLM}(h_l|\gamma_l, \beta_l) =$$
$$\textit{SFiLM}\big(D(g)h_l|D(g)\gamma_l, D(g)\beta_l\big), g \in \mathrm{SO}(3) \quad (6)$$

The proposition is proved by the orthogonal property of the Wigner D-matrices $D$ and Schur's lemma (Schur, 1905), please see Appendix A.2 for details.

## 5. Experiments

### 5.1. Simulation Experiments

**Experimental Settings**  We conduct simulation experiments using the MimicGen (Mandlekar et al., 2023) environment, built on the Mujoco simulator (Todorov et al., 2012), which features diverse tasks that are contact-rich, precise, and long-horizon (see Figure 4). Unlike scripted demonstrations, which are unimodal, or Reinforcement Learning (RL) agent-generated demonstrations, which are Markovian, MimicGen generates multi-modal, non-Markovian trajectories from a few human demonstrations (Mandlekar et al., 2021), making it well-suited for benchmarking learning from human demonstrations.

MimicGen provides observations in the form of RGBD images from both a front view and an in-hand view, along with a 7-DoF robot state. The RGB images have a resolution of $84 \times 84 \times 3$, while RGBD data can be used to reconstruct either 3D colored voxels ($84^3$) or colored point clouds (PCD) with 1024 points. For PCD, we exclude table points, following DP3 (Ze et al., 2024). The action space in MimicGen consists of a 6-DoF gripper pose and a 1-DoF gripper aperture. Three control modes are used: Absolute Control, which defines the gripper trajectory in the robot frame; Relative Control, which defines it in the current gripper frame; and Velocity Control, which determines the next gripper pose relative to the previous one (Chi et al., 2024).

To evaluate robustness, we modify four MimicGen tasks with $\mathrm{SE}(3)$ initialization by randomly tilting the table within a defined range and randomly placing objects on the tabletop while keeping the robot base upright. Benchmarking is conducted across three difficulty levels with progressively increasing tilt ranges: $[0]$, $[-15°, 15°]$, and $[-30°, 30°]$. Additionally, we also compare various baselines across all 12 original MimicGen tasks.

We compare several baselines in our experiments: 1) EquiDiff (Wang et al., 2024b) – an SO(2)-equivariant diffusion policy using either voxel or RGB image observations. 2) DiffPo (Chi et al., 2023) – the original diffusion policy, employing either a convolutional (-C) or transformer (-T) backbone in the diffusion network. 3) EquiBot (Yang et al., 2024a) – an SO(3)-equivariant diffusion policy with up to degree $l = 1$ representations. 4) DP3 (Ze et al., 2024) – a diffusion policy based on point-cloud representations. 5) ACT (Zhao et al., 2023b) (Action Chunking Transformer) – a model capturing multi-modality via a Variational Autoencoder (VAE). 6) BC-RNN (Mandlekar et al., 2021) – a behavioral cloning approach that captures multi-modality using a Gaussian Mixture Model (GMM) and accounts for non-Markovian dynamics via a Recurrent Neural Network (RNN). A relevant baseline, ET-SEED (Tie et al., 2024), is not included because the code was unavailable before the initial submission. Following (Chi et al., 2023; Wang et al., 2024b), we train all baselines using DDPM (Ho et al., 2020) with 100 denoising steps. For details on hyperparameters, see Appendix D. We report the maximum test success rate throughout training, averaging results over 50 rollouts for each of the three seeds.

**Results on Tasks with** $\mathrm{SE}(3)$ **Initialization**  Table 1 shows that SDP outperforms all baselines across all tilting ranges, except for the Coffee $0°$ task, demonstrating superior sample efficiency. Notably, as the tilting range increases, SDP achieves a more significant relative performance improvement over the baselines. This highlights SDP's strong $\mathrm{SE}(3)$ generalization, enabled by its continuous $\mathrm{SE}(3)$ equivariance property. However, performance declines for all methods, including SDP, as the tilting range increases. We hypothesize that this drop is caused by point-cloud occlusion and object instability due to gravity, both of which disrupt $\mathrm{SE}(3)$ equivariance.

**Results on Tasks with** $\mathrm{SE}(2)$ **Initialization**  Table 2 shows that SDP outperforms all baselines across 10 tasks, except for Coffee and Coffee_Preparation. Despite the variations in $\mathrm{SE}(2)$, SDP still demonstrates a notable advantage, suggesting that its continuous $\mathrm{SE}(2)$ equivariance benefits learning more effectively than the discrete $C_8$ equivariance in EquiDiff. The lower performance of SDP on Coffee and Coffee_Preparation may be attributed to the low-resolution point clouds, which struggle to capture fine details—such as the slack between the coffee pod and its receptacle—potentially hindering precise manipulation.

### 5.2. Physical Experiments

**Experimental Settings**  We further evaluate the performance of SDP across 5 physical tasks in Figure 5, using a robot station shown in Figure A1. Turn_Lever involves manipulating an articulated object, while Push_Eraser requires pushing a small eraser. Grasp_Box challenges the policy to maintain a closed kinematic chain. Flip_Book involves rich contact between the end-effector, the tabletop, and the book. Pack_Package is a long horizon task. The observations are captured by two stationary RGBD cameras positioned above the workspace to minimize occlusion. Point clouds with 1024 points are reconstructed from the RGBD images (for

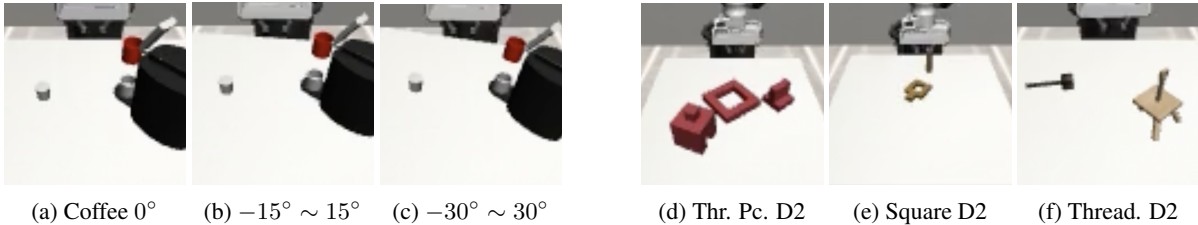

(a) Coffee $0°$ (b) $-15° \sim 15°$ (c) $-30° \sim 30°$ (d) Thr. Pc. D2 (e) Square D2 (f) Thread. D2

*Figure 4.* MimicGen tasks with $SE(3)$ initialization ((a)-(c), showing 1 of 4 tasks) and $SE(2)$ initialization ((d)-(f), showing 3 of 12 tasks).

*Table 1.* Evaluation success rate on 4 MimicGen tasks with 3 levels of $SE(3)$ initialization. We train all the baselines on progressively tilted environments with 100 demonstrations. As the degress of $SE(3)$ initialization increases, SDP maintains reasonable performance while the performance of other baselines drop severely. Results averaged over three seeds.

| Method | Equ | Ctrl | Obs | Coffee 0° | Coffee 15° | Coffee 30° | Three Pc. Assembly 0° | Three Pc. Assembly 15° | Three Pc. Assembly 30° | Square 0° | Square 15° | Square 30° | Threading 0° | Threading 15° | Threading 30° | Average 0° | Average 15° | Average 30° |
|---|---|---|---|---|---|---|---|---|---|---|---|---|---|---|---|---|---|---|
| SDP | $SE(3)$ | Rel | PCD | 63 | **54** | 33 | 67 | **49** | 37 | 62 | **38** | 31 | 60 | 53 | 39 | 63 | 49 | 35 |
| EquiDiff (Wang et al., 2024b) | $C8 \subset SO(2)$ | Abs | Voxel | **65** | 43 | 29 | 37 | 15 | 8 | 39 | 3 | 3 | 39 | 20 | 10 | 45 | 20 | 13 |
| DiffPo (Chi et al., 2023) | N/A | Abs | RGB | 44 | 23 | 16 | 4 | 2 | 2 | 8 | 0 | 1 | 17 | 10 | 8 | 18 | 9 | 7 |
| EquiBot (Yang et al., 2024a) | $SO(3)$ | Abs | PCD | 0 | 1 | 0 | 1 | 1 | 1 | 0 | 1 | 1 | 6 | 4 | 0 | 2 | 2 | 1 |

*Table 2.* Evaluation success rate on 12 MimicGen tasks with $SE(2)$ initialization. We train all the baselines with 100 demonstrations. SDP demonstrates the best performance on 10 tasks. Results averaged over three seeds.

| Method | Ctrl | Obs | Stack D1 | Stack Three D1 | Square D2 | Threading D2 | Coffee D2 | Three Pc. Assembly D2 | Hammer Cleanup D1 | Mug Cleanup D1 | Kitchen D1 | Nut Assembly D0 | Pick Place D0 | Coffee Preparation D1 | Average Success Rate |
|---|---|---|---|---|---|---|---|---|---|---|---|---|---|---|---|
| SDP | Rel | PCD | **100** | **98** | **62** | **60** | 63 | **67** | **82** | **54** | **89** | **92** | **73** | 73 | **76** |
| EquiDiff (Wang et al., 2024b) | | Voxel | 99 | 75 | 39 | 39 | **65** | 37 | 70 | 53 | 85 | 67 | 58 | **80** | 64 |
| EquiDiff (Wang et al., 2024b) | | RGB | 93 | 55 | 25 | 22 | 60 | 15 | 65 | 49 | 67 | 74 | 42 | 77 | 54 |
| DiffPo-C (Chi et al., 2023) | Abs | RGB | 76 | 38 | 8 | 17 | 44 | 4 | 52 | 43 | 67 | 55 | 35 | 65 | 42 |
| DiffPo-T (Chi et al., 2023) | | RGB | 51 | 17 | 5 | 11 | 47 | 1 | 48 | 30 | 54 | 31 | 15 | 38 | 29 |
| DP3 (Ze et al., 2024) | | PCD | 69 | 7 | 7 | 12 | 34 | 0 | 54 | 21 | 45 | 16 | 12 | 10 | 24 |
| ACT (Zhao et al., 2023b) | | RGB | 35 | 6 | 6 | 10 | 19 | 0 | 38 | 23 | 37 | 42 | 7 | 32 | 21 |
| EquiDiff (Wang et al., 2024b) | | Voxel | 95 | 59 | 25 | 33 | 55 | 5 | 64 | 39 | 69 | 53 | 40 | 48 | 49 |
| EquiDiff (Wang et al., 2024b) | Vel | RGB | 75 | 25 | 11 | 11 | 41 | 1 | 49 | 29 | 61 | 44 | 29 | 49 | 35 |
| DiffPo-C (Chi et al., 2023) | | RGB | 81 | 26 | 6 | 13 | 43 | 2 | 43 | 25 | 42 | 42 | 35 | 42 | 33 |
| BC RNN (Mandlekar et al., 2021) | | RGB | 59 | 12 | 8 | 7 | 37 | 0 | 32 | 19 | 31 | 35 | 21 | 14 | 23 |

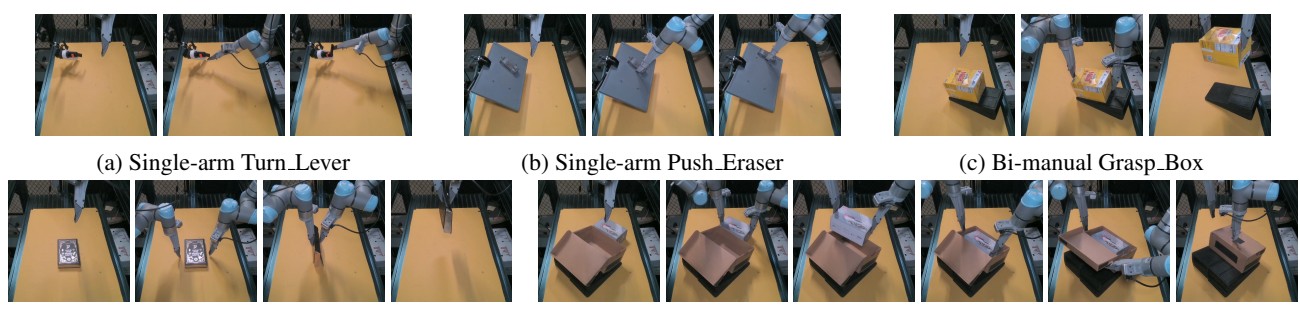

(a) Single-arm Turn_Lever (b) Single-arm Push_Eraser (c) Bi-manual Grasp_Box

(d) Bi-manual Multi-step Flip_Book (e) Bi-manual Multi-step Pack_Package

*Figure 5.* Five physical robotic manipulation tasks.

Push_Eraser we also tested 2048 points). The actions are the 6 DoF gripper poses for single-arm tasks or 12 DoF gripper poses for bi-manual tasks.

**Training Dataset from Human Demonstrations** We use Gello (Wu et al., 2023) to collect demonstrations with objects initialized in random $SE(3)$ poses. For the single-arm tasks, we collect 30 successful human demonstrations. Additionally, we record an extra 10% demos as the recov-

*Table 3.* Success rate (%) of 5 physical experiments over 20 evaluation episodes. The action space and number of training demonstrations are listed under each task. Overall, SDP is 61% better than EquiDiff and 71% better than DiffPo-C. Results from one seed. * using point clouds with 2048 points.

| Method | Turn Lever 6 DoF 33 Demos | Push Eraser 6 DoF 33 Demos | Grasp Box 12 DoF 33 Demos | Flip Book 12 DoF 66 Demos | Pack Package 12 DoF 66 Demos | Avg. Succ. Rate |
|---|---|---|---|---|---|---|
| SDP | 80 | 35/ 90* | 85 | 65 | 70 | 78 |
| EquiDiff | 20 | 30 | 35 | 0 | 0 | 17 |
| DiffPo-C | 10 | 10 | 15 | 0 | 0 | 7 |

ery demos at specific poses. Similarly, for the bi-manual Grasp_Box task, we collect 33 demos at random $SE(3)$ poses. For more challenging bi-manual tasks like Flip_Book and Pack_Package, we collect 66 demos. Figure A3 visualizes the $SE(3)$ training pose distribution for all five tasks. Further details on the tasks are provided in Appendix B.2.

**Results and Discussion**  We benchmark SDP against the top two baselines, EquiDiff (Wang et al., 2024b) and DiffPo-C (Chi et al., 2023), from the simulation experiment (Table 1). We train all three models using DDIM (Song et al., 2021) and inference with 8 denoising steps for all tasks. Each baseline is evaluated on 20 rollouts per physical task, with each rollout initialized using object poses in novel $SE(3)$ poses unseen in the training set. The success rates are summarized in Table 3, with a detailed breakdown of the success rates provided in Table A1. For the Push_Eraser task, increasing the PCD resolution to 2048 points enables accurate localization of the eraser, resulting in a performance improvement of over 50%.

SDP significantly outperforms all baseline methods across every task and embodiments, achieving a 61% higher success rate than EquiDiff and a 71% improvement over DiffPo-C. These significant gains in sample efficiency and spatial generalization are largely attributed to its inherent $SE(3)$ equivariance. For instance, in the Turn_Lever task, SDP successfully locates and rotates a lever that is randomly clamped in 3D space. In comparison, EquiDiff frequently misdirects the gripper to the workspace center, entirely missing the lever, while DiffPo approaches the lever but only hovers nearby without engaging it. Further discussion on common failures can be found in Appendix B.3.

### 5.3. Ablation Study

Table 4 presents six ablations: 1) **Discrete SDP** - replaces SDTU that has continuous equivariance with discrete equivariant denoising U-net with Octahedron (cubical) discretization, this is a $SE(3)$ adaption of (Wang et al., 2024b; Zhao et al., 2025). 2) **SDP Absolute Action** – defines actions in the workspace frame instead of SDP's relative action formu-

lation, which defines actions in the current gripper frame. 3) **DP3-canonical** - DP3 (Ke et al., 2024) with canonicalized observation-action space, by transforming the point cloud and trajectory to the gripper frame. This achieves $SE(3)$-invariance. 4) **EquiBot Rel.** – removes SDP's spherical Fourier features and replaces its model with EquiBot (Yang et al., 2024a), while keeping the relative action formulation. 5) **DP3 Absolute Action** - (Ke et al., 2024) the original DP3 policy, this baseline ablates relative action, spherical representation, and equivariance. 6) **EquiBot Absolute Action** – removes both the spherical Fourier features and the relative action formulation, adopting EquiBot (Yang et al., 2024a) in the absolute action formulation.

The results are shown in Table 4. **Discrete SDP** trivially modifying (Wang et al., 2024b) to be $SE(3)$ equivariant, sufferers from discretization error thus underperforms SDP. The relative action formulation that achieves 3D translational equivariance, also plays a key role, as its removal in **SDP Abs.** causes major performance drops, particularly in the Coffee and Square tasks. **DP3-canonical** leverages $SE(3)$ invariant features, outperforms **DP3** that did not leverage, but still underperforms SDP by a large margin, demonstrating the advantage of equivariant features. This matches the finding in (Miller et al., 2020). The **EquiBot Rel.** ablations result in significant performance drops across all four tasks, indicating that the spherical Fourier representation is the most critical factor for SDP's performance. Finally, **DP3 Abs., EquiBot Abs.** further amplifies the performance degradation, demonstrating that removing both the relative action formulation and spherical Fourier representation is more detrimental than removing either one alone.

*Table 4.* Ablation study. The relative action space and the spherical representation are critical for the $SE(3)$ generalization, while the latter one is more important. Results from one seed.

| Method | Rel. Act. | Spher. Rep. | Equi-variance | Coffee 15° | Thr. Pc. As. 15° | Square 15° | Thread. 15° | Avg. SR |
|---|---|---|---|---|---|---|---|---|
| SDP | ✓ | ✓ | $SE(3)$ equ. | 54 | 49 | 38 | 53 | 49 |
| Discrete SDP | ✓ | ✓ | Octahedron | 42 | 16 | 34 | 48 | 35 |
| SDP Abs. | ✗ | ✓ | $SO(3)$ equ. | 18 | 42 | 0 | 44 | 26 |
| DP3-canonical | ✓ | ✗ | $SE(3)$ inv. | 40 | 0 | 8 | 12 | 15 |
| EquiBot Rel. | ✓ | ✗ | $SE(3)$ equ. | 18 | 2 | 4 | 6 | 8 |
| DP3 Abs. | ✗ | ✗ | None | 20 | 0 | 0 | 4 | 6 |
| EquiBot Abs. | ✗ | ✗ | $SO(3)$ equ. | 1 | 1 | 1 | 4 | 2 |

### 5.4. Additional Studies

**Performance VS Degree $l$ in spherical Fourier features** As shown in Table 5, increasing the degree from 1 to 2 improves performance across all tasks. However, beyond degree 2, performance saturates while computational cost increases. Since SDP is designed for real-time robot control, we select $l = 2$.

*Table 5.* Success rate VS degree $l$ of the spherical Fourier feature. Results from one seed.

| Degree $l$ | Coffee 15° | Thr. Pc. As. 15° | Square 15° | Threading 15° | Avg. SR |
|---|---|---|---|---|---|
| 3 | 52 | 52 | 66 | 52 | 56 |
| 2 (SDP) | 54 | 49 | 38 | 53 | 49 |
| 1 | 44 | 4 | 10 | 46 | 26 |

**Sample Efficiency**   To assess the impact of training dataset size on performance, we evaluate SDP, EquiDiff, and DiffPo on four MimicGen tasks with tilted angles in the range $[0, 15°]$, using 100, 316, or 1000 demonstrations. Figure 6 summarizes the results, with each point representing the average success rate across all four tasks.

SDP achieves a success rate of 48.5% using only 100 demonstrations, surpassing EquiDiff by 4% while utilizing just one-tenth of the data, indicating a $10\times$ improvement in data efficiency. Similarly, EquiDiff attains a 20% success rate with 100 demonstrations, matching the performance of DiffPo, which requires approximately 300 demonstrations, supporting a $3\times$ gain in data efficiency.

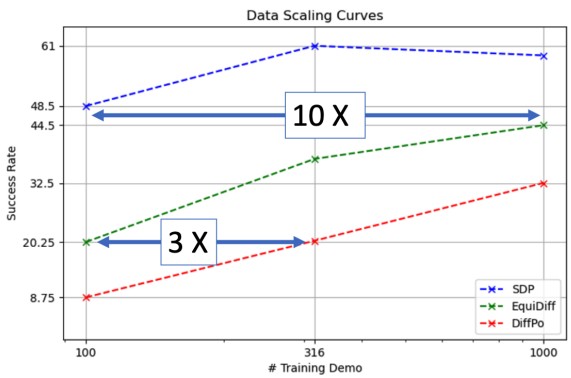

*Figure 6.* Impact of training dataset size on task success rates: increasing the number of demonstrations from 100 to 316 (a $3\times$ increase) yields an average success rate improvement of 9–12% across four tasks, with initial success rates of 48.5% (SDP), 20.3% (EquiDiff), and 8.8% (DiffPo). At 1,000 demonstrations ($10\times$), the average success rate saturates for SDP at 60%, while EquiDiff and DiffPo continue to improve, reaching 44.5% and 32.5%, respectively.

The observed performance saturation of SDP beyond 300 demonstrations may be attributed to scene occlusions, as relying solely on agent-view and in-hand cameras can obscure critical areas of the workspace. Additionally, all demonstrations are generated from 10 raw human demonstrations (Mandlekar et al., 2023), so increasing the number of generated demonstrations may not increase diversity in the data. Furthermore, the kinematic constraints of the robot may limit its ability to access certain regions, thereby impacting overall task success.

**Inference Speed**   Table 6 presents the inference time comparing SDP with four other baselines. DiffPo is the fastest (0.09s) while ETSEED is the slowest (29.4s). The inference time of SDP is on the same order of magnitude as that of the best baseline-DiffPo (approximately $5\times$), while SDP achieves continuous $SE(3)$ equivariance, significantly better performance than DiffPo and EquiDiff in simulation and physical experiments, and does not require preprocessing.

SDP leverages spherical features by using EquiformerV2 (Liao et al., 2024) and SDTU, which is more lightweight ($66\times$ faster inference speed, $32\times$ larger batch size) than the $SE(3)$-transformer that ET-SEED is based on. Moreover, SDP supports high orders of spherical harmonics, which is more expressive than Vector Neuron. Lastly, SDP achieves continuous $SE(3)$-equivariance, where EquiDiff enforces discrete $C8 \subset SO(2)$ equivariance.

*Table 6.* Comparison of inference time. At the costs of $5\times$ solver than DiffPo, SDP achieves continuous $SE(3)$ equivariance and does not need preprocessing. The inference time and the training batch size are tested on a commercial GPU with 24GB RAM.

| Method | Inference Time (s) ↓ | Pre-processing | Batch Size ↑ | Equivariance | Neural Network |
|---|---|---|---|---|---|
| SDP | 0.44 | No | 32 | $SE(3)$ | EquiformerV2, SDTU |
| DiffPo | 0.09 | No | 64 | None | Convolution |
| EquiDiff | 0.14 | No | 64 | $C8 \subset SO(2)$ | ESCNN |
| EquiBot | 0.18 | Segmentation | 64 | $SE(3)$ | Vector Neuron |
| ET-SEED | 29.4 | Segmentation | 1 | $SE(3)$ | $SE(3)$-Transformer |

## 6. Conclusion and Limitations

This paper introduces the Spherical Diffusion Policy (SDP), an $SE(3)$-equivariant policy that generalizes to 3D scene arrangements using only a few demonstrations. SDP achieves this through three key components: 1) spherical Fourier features, providing compact and precise representations for continuous $SO(3)$ equivariance; 2) spherical FiLM, enforcing equivariant conditioning; and 3) a spherical denoising temporal U-net, ensuring spatiotemporal equivariant denoising. SDP significantly outperforms strong baselines across simulation and real-world experiments, demonstrating effectiveness in both single-arm and bi-manual embodiments.

One limitation of the proposed method is that it operates in position control, ignoring contact forces, which leads to protective stops in the Flip_Book task. An important future direction is to address this by learning a compliant, force-aware policy (Kohler et al., 2024; Hou et al., 2024) in an equivariant manner. Another limitation is the low-resolution point cloud processing in the observation encoder, which struggles to capture fine details, such as these in the Push_Eraser task. Using a more efficient Graph Neural Network (Zhao et al., 2021; Luo et al., 2024) could help mitigate this issue.

## Impact Statement

On the bright side, the proposed method enhances spatial generalization for manipulation policies, making it potentially deployable in real-world scenarios to significantly reduce human workload. On the dark side, the method lacks awareness of common scenes, which could result in risky actions such as harming individuals, causing fires, or damaging objects.

## Acknowledgments

The authors would like to thank Michael Schultz and Nathan Gere for the help on the physical experiments, Haojie Huang for the help on the simulation environments, Pranay Thangeda and Erica Aduh for the help on the robot platform setups.

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

# A. Proofs

## A.1. Proof of Proposition 4.1

*Proof.* Focusing on the right-hand side of the Equation 4:

$$\sum_{j \in T} \sum_{i \in in} D^l_{mn}(g) h^i_{l,m,j} w^{i,o}_{l,j-t} \tag{7}$$

$$= D^l_{mn}(g) \sum_{j \in T} \sum_i h^i_{l,m,j} w^{i,o}_{l,j-t} \tag{8}$$

$$= D^l_{mn}(g) h^o_{l,m,t}, \tag{9}$$

the line 8 is because of Schur's lemma (Schur, 1905), which states that any linear operation of SO(3) irreps acts as on each irreducible subspace is equivariant. □

## A.2. Proof of Proposition 4.2

*Proof.* Focusing on the right-hand side of Equation 6:

$$\text{SFiLM}\big(D(g)h_l | D(g)\gamma_l, D(g)\beta_l\big) \tag{10}$$

$$= (D(g)\gamma_l)^{\mathrm{T}} D(g) h_l \frac{D(g)h_l}{||D(g)h_l||} + D(g)\beta_l \tag{11}$$

$$= \gamma_l^{\mathrm{T}} D(g)^{\mathrm{T}} D(g) h_l \frac{D(g)h_l}{||D(g)h_l||} + D(g)\beta_l \tag{12}$$

$$\tag{13}$$

Because the Wigner D-matrices are orthogonal, we have:

$$\text{SFiLM}\big(D(g)h_l | D(g)\gamma_l, D(g)\beta_l\big) \tag{14}$$

$$= \gamma_l^{\mathrm{T}} h_l \frac{D(g)h_l}{||h_l||} + D(g)\beta_l \tag{15}$$

$$= D(g)(\gamma_l^{\mathrm{T}} h_l \frac{h_l}{||h_l||} + \beta_l) \tag{16}$$

$$= D(g) \cdot \text{SFiLM}(h_l | \gamma_l, \beta_l), \tag{17}$$

the line 16 is based on Schur's lemma (Schur, 1905). □

# B. Physical Experiments and Results

## B.1. Physical Experiment Workstation

Our physical robotic workstation, as shown in Figure A1, is composed of two collaborative UR5e manipulators, each equipped with a compliant ray-fin finger(Crooks et al., 2016).Two scene cameras (RealSense D415) are positioned on either side of the workspace. GELLO controllers (Wu et al., 2023) are utilized to collect human demonstrations for physical robotic manipulation tasks.

## B.2. Physical Manipulation Tasks and Training Datasets

We experiment with SDP on five physical tasks, as shown in Figure A2 : (a) Turn_Lever involves manipulating an articulated object and (b) Push_Eraser requires pushing a small object with a single manipulator; (c) Bi-manual Grasp_Box challenges the policy to maintain a closed kinematic chain; (d) Flip_Book involves rich contact between the end-effector and the book while transforming book's pose with dexterous complex coordinated manipulation; (e) Pack_Package is a long horizon task.

**Turn_Lever**: An expert demonstrator moves one ray-fin finger to make flush contact with the edge of the lever and then turn the Lever counter-clockwise at least 60 degree around the fulcrum. Otherwise the task has failed. The lever, initialized with

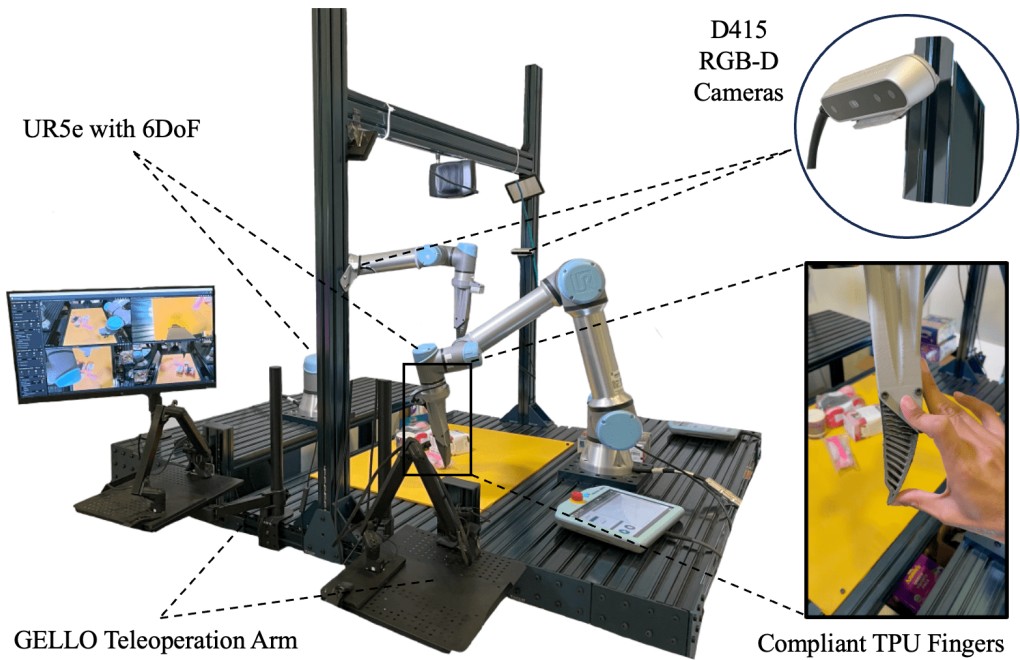

*Figure A1.* Overview of the manipulation experimental setup: two UR5e manipulators, each with a compliant ray-fin fingers, two stationary overhead cameras, and GELLO teleop controllers.

a $SE(3)$ pose, is flexibly positioned with a clamp, using a combination of pitch and yaw angles within the 3D workspace of the manipulator. A total of 33 human demonstrations have been collected, composed of 30 successful demo and 3 (10%) recovery demonstrations where failed states were corrected to reach to the successful goal state.

**Push_Eraser**: An expert demonstrator moves one ray-fin finger to make contact with the Eraser, which is initialized with an $SE(3)$ pose within the marked boundary on the whiteboard. Once the contact is secure, the demonstrator will move the ray-fin finger to push the Eraser, in a straight line, towards the closest edge, until the eraser is outside the marked rectangle, achieving a successful goal state. Otherwise the task has failed. The whiteboard is positioned, flexibly with a clamp, with approximate pitch angles of -15, 0, 30, 45, 60, and 90 degrees and approximate yaw angles of -15, 0, and 15 degrees within the 3D workspace of the manipulator. In total, 33 human demonstrations were collected, consisting of 30 successful demonstrations and 3 (10%) recovery demonstrations.

For the bi-manual manipulation tasks, we design a set of 432 distinct $SE(3)$ poses, consisting of 9 regions on the x-y plane, 3 discrete pitch angles (0°, 8°, or 16°) provided by an 8° wedge, and 16 discrete yaw angles with 15° or 30° increments. We randomly sample an $SE(3)$ pose from this set to position a box, book, or container for collecting human demonstrations.

**Grasp_Box**: an expert demonstrator moves two ray-fin fingers to pinch grasp the box at a sampled $SE(3)$ pose, then lifts the pinched box minimally 40cm above the flat surface. Otherwise the task has failed. Similarly, we collect 33 human demonstrations including 3 recovery demos.

**Flip_Book**: an expert demonstrator moves two ray-fin fingers to pinch grasp the book along its medium dimension at a sampled $SE(3)$ pose, then rotates the pinched book in-hand so that the book is pinched along its smallest dimension, and then lifts the book minimally 40cm above the flat surface. Otherwise the task has failed. Due to the precision required for coordinated finger movements, we collect 66 human demonstrations including 6 recovery demos.

**Pack_Package**: An expert demonstrator moves two ray-fin fingers to pinch-grasp the box, transports it to the pre-pack pose above the container, places the box inside, moves to the pre-lid-close pose, and finally closes the lid. The final goal state of the task is the box inside the container with the lid closed. If any step fails, the task is considered a failure. Due to the precision required for coordinated finger movements, we collect 66 human demonstrations, including 6 recovery demonstrations.

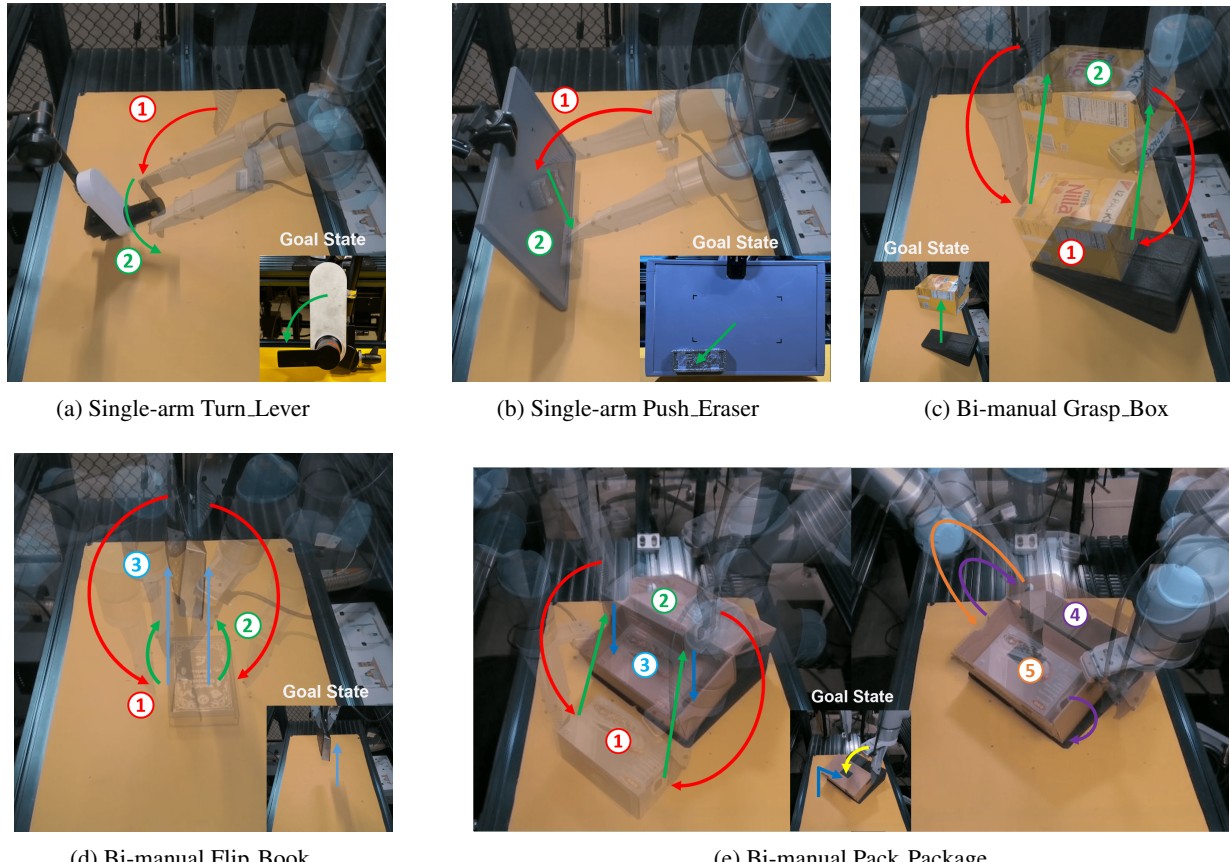

(a) Single-arm Turn_Lever    (b) Single-arm Push_Eraser    (c) Bi-manual Grasp_Box

(d) Bi-manual Flip_Book    (e) Bi-manual Pack_Package

*Figure A2.* Demonstrated robot actions for each task: a) Turn_Lever, b) Push_Eraser, and c) Grasp_Box, each task with two trajectory segments and their respective goal states; d) Flip_Book task with three trajectory segments and goal states, where both manipulators must perform coordinated movement after the initial pinch ; e) Pack_Package with five trajectory segments and goal states.

## B.3. Detailed Evaluation Results

We evaluate the baseline performance with 20 rollouts, using object poses randomly sampled from the SE(3) pose set in training. All poses are annotated, and evaluation is conducted on novel, unseen poses.

For all five physical tasks, we report the success rate for each intermediate goal state in Table A1. We compare SDP with EquiDiff (Wang et al., 2024b) and DiffPo (Chi et al., 2023). SDP achieves a strong performance with a minimum 90% success rate on the first step, where both EquiDiff and DiffPo perform poorly. For subsequent steps, the success rate drops by up to 20% in the Flip_Book task, where precise coordination of both fingers is required for the in-hand pose rotation.

Figure A4 provides examples of task successes and failures. The most common failure cases for SDP occur during the book flip step. Lack of precise coordination between the two fingers may cause the book to either drop or be pinched too tightly, leading to a robot fault. Other failures include invalid pinching during the pick step or object drops due to a loose grip. For the Pack_Box task, collisions with the container may occur during the transfer of the pinched box, and misalignment or incorrect placement can lead to collisions during the packing step. Interestingly, when positioning the lid to close, the robot may mistakenly identify the object as the lid. For the Turn_Lever task, the finger may drift away from the lever while attempting to complete the required rotation. For the Push_Eraser task, the robot pushes in the wrong direction, failing to push the eraser across the boundary. For the Grasp_Box task, SDP struggles to pinch the box when its long dimension is parallel to the robot's front (i.e., when the box has a yaw angle of 0 or 180 degrees).

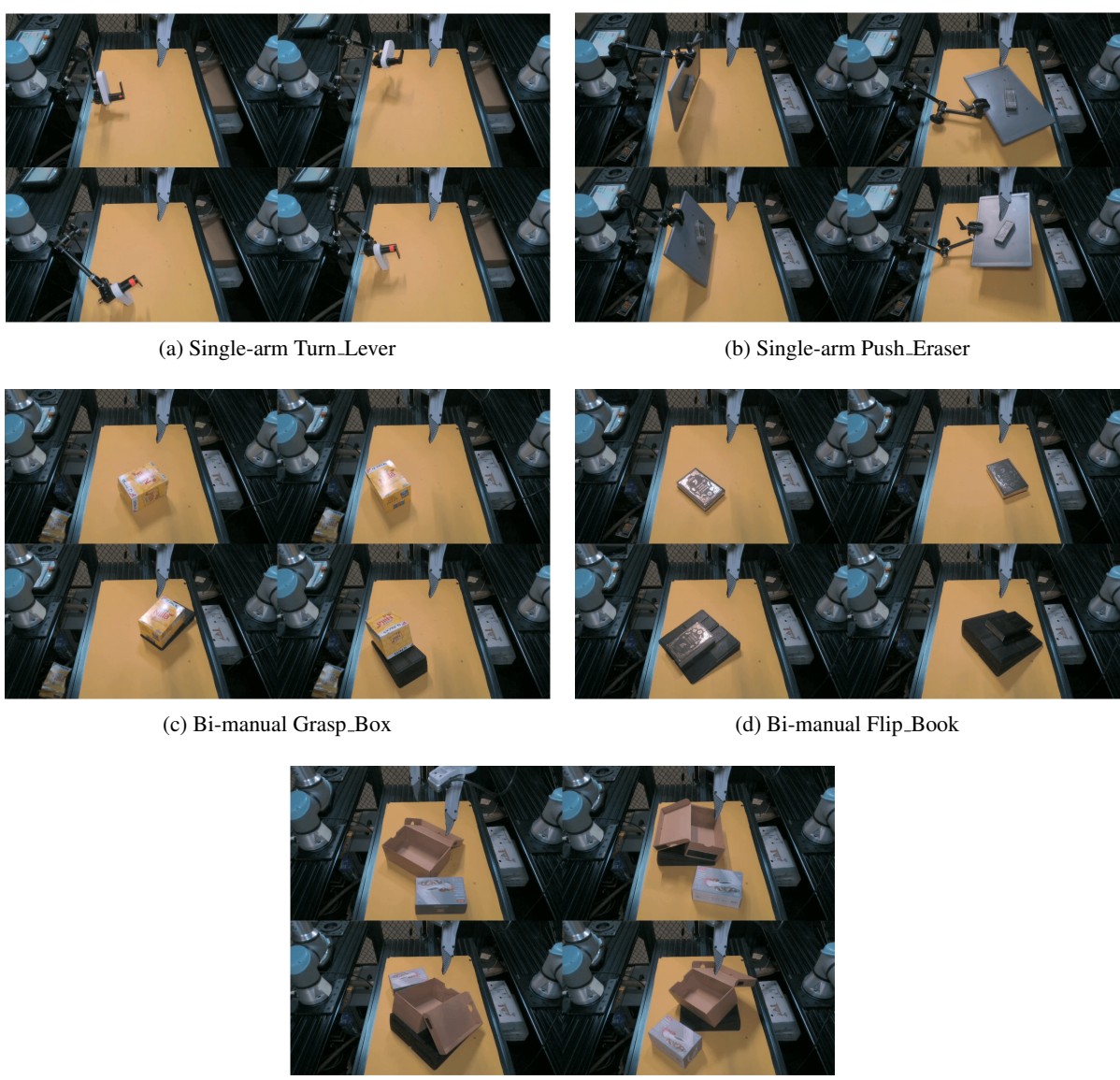

(a) Single-arm Turn_Lever

(b) Single-arm Push_Eraser

(c) Bi-manual Grasp_Box

(d) Bi-manual Flip_Book

(e) Bi-manual Pack_Package

*Figure A3.* Visualization of SE(3) Pose Distribution for Five Physical Tasks. The initial state of 4 out of 20 episodes are visualized.

## C. Additional Background

### C.1. Equivariant Diffusion

The theory of Equivariant Diffusion has been extensively investigated in (Köhler et al., 2020; Brehmer et al., 2024; Ryu et al., 2024; Wang et al., 2024b). Based on these works, we summarize the equivariance property of the policy and the denoising function of the policy for completeness. There are two scenarios: diffuse equivariance and denoise equivariance.

**Proposition C.1** (Diffuse equivariance)**.** *If the policy is equivariant to group $G$, i.e., $\pi(gS) = g\pi(S)$, and the distribution $\mathcal{D}$ from which to sample the noise, is invariant to group $G$, i.e., $\mathcal{D} = g\mathcal{D}$, and the the denoising function satisfies*

$$\epsilon_\theta(S, \pi(S) + \epsilon, k) = \epsilon, \quad \epsilon \sim \mathcal{D}, \tag{18}$$

*then the denoising function is equivariant to group $G$, i.e., $\epsilon(gS, gA^k, k) = g\epsilon(S, A^k, k)$,*

*Proof.* We assume the denoising function satisfies Equation 18. Since the equation holds for all $\epsilon \sim \mathcal{D}$ and $\mathcal{D}$ is $G$-invariant,

*Table A1.* Breakdown of success rates at each step for five physical experiments over 20 evaluation episodes. The action space and number of training demonstrations are same as in Table 3.

| Method | Turn Lever | | Push Eraser | | Grasp Box | | Flip Book | | | Pack Box | | | | |
| --- | --- | --- | --- | --- | --- | --- | --- | --- | --- | --- | --- | --- | --- | --- |
| | Contact | Rotate | Contact | Push | Grasp | Lift | Grasp | Flip | Lift | Pick | Transport | Pack | Locate | Close |
| SDP | 90 | 80 | 90 | 90 | 90 | 85 | 100 | 80 | 65 | 95 | 85 | 75 | 70 | 70 |
| EDP | 35 | 20 | 40 | 30 | 45 | 35 | 0 | 0 | 0 | 45 | 35 | 10 | 0 | 0 |
| DP | 30 | 10 | 15 | 10 | 30 | 15 | 35 | 0 | 0 | 70 | 35 | 5 | 5 | 0 |

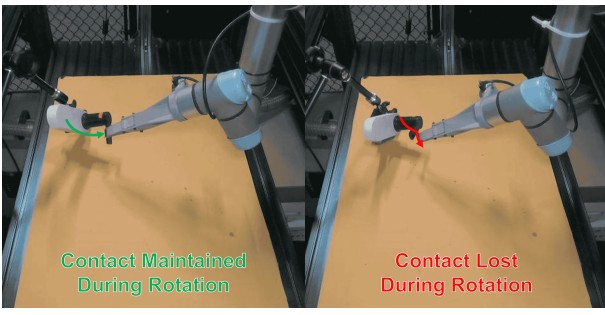

(a) Single-arm Turn_Lever  (b) Single-arm Push_Eraser

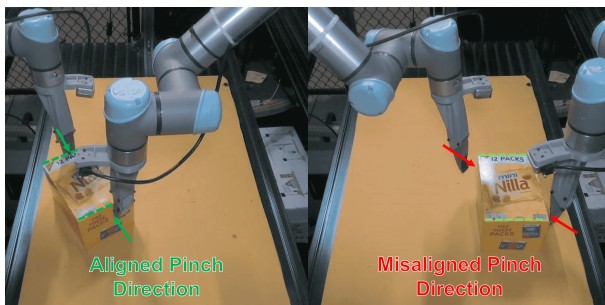

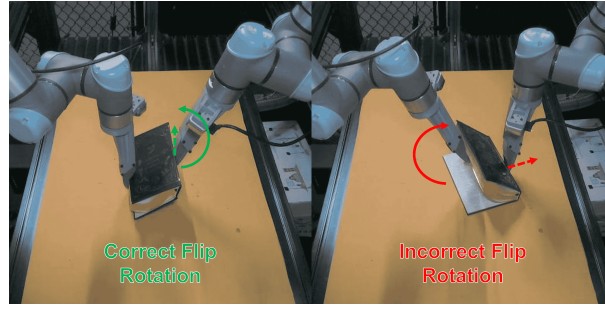

(c) Bi-manual Grasp_Box  (d) Bi-manual Flip_Book

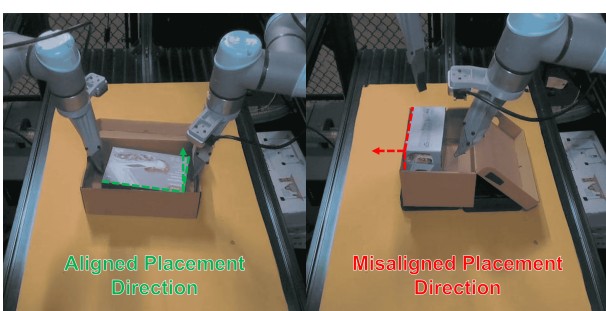

(e) Bi-manual Pack_Package

*Figure A4.* Examples of successes and failures for each task, with green indicating successful behaviors and red indicating failures.

we can evaluate at $gS$ and $g\epsilon$,

$$\epsilon_\theta(gS, \pi(gS) + g\epsilon, k) = g\epsilon.$$

Using the equivariance of $\pi$ and substituting in Eqn.18 for $\epsilon$ gives

$$\epsilon_\theta(gS, g\pi(S) + g\epsilon, k) = g\epsilon_\theta(S, \pi(S) + \epsilon, k)$$

as desired. $\qquad \square$

**Proposition C.2** (Denoise equivariance). *The action prediction is equivariant to group $G$, i.e., $\pi(gS) = g\pi(S)$, when the denoising function is equivariant to group $G$, i.e., $\epsilon(gS, gA^k, k) = g\epsilon(S, A^k, k)$, and the distribution $\mathcal{D}$ from which to sample the noise, is invariant to group $G$, i.e., $\mathcal{D} = g\mathcal{D}$.*

*Proof.* Simplifying Equation. 1 by dropping $t$ we have:

$$A^{k-1} = \alpha\big(A^k - \gamma\epsilon_\theta(S, A^k, k) + z\big), z \sim \mathcal{D} \tag{19}$$

When $k = K$, the denoise is sampled form $\mathcal{D}$: $\epsilon_\theta(S, A, K) = d_1, d_1 \sim \mathcal{D}$, and the denoised action $A^{K-1}$ is:

$$
\begin{align}
A^{K-1} &= \alpha(0 - \gamma d_1 + d_2), \quad d_1, d_2 \sim \mathcal{D} \tag{20}\\
&= \alpha(d_2 - \gamma d_1) \tag{21}\\
&= \alpha(g d_2' - \gamma g d_1'), \quad d_1', d_2' \sim \mathcal{D} \tag{22}\\
&= g\alpha(d_2' - \gamma d_1') \tag{23}\\
&= gA^{K-1} \tag{24}
\end{align}
$$

When $k = K - 1$, the denoise is applied to the noisy action, to generate the cleaner action. Transforming the input to the denoising function in Equation. 19 by $g$:

$$
\begin{align}
\alpha\big(A^{K-1} &- \gamma\epsilon_\theta(gS, gA^{K-1}, K-1) + z\big) \tag{25}\\
&= \alpha\big(gA^{K-1} - g\gamma\epsilon_\theta(S, A^{K-1}, K-1) + gz\big) \tag{26}\\
&= g\alpha\big(A^{K-1} - \gamma\epsilon_\theta(S, A^{K-1}, K-1) + z\big) \tag{27}\\
&= gA^{K-2} \tag{28}
\end{align}
$$

Equation. 28 holds for $k = K - 1, K - 2, ..., 1$, thus by applying it iteratively, we have $gA^0$. $\qquad \square$

Proposition. C.1, C.2 specify the prerequisites of an equivariant diffusion policy. Empirically we find that the group invariant distribution $\mathcal{D}$ can be relaxed to a Gaussian distribution $\mathcal{N}$, which is simple and achieves good performance.

### C.2. Translation Invariance by Canonicalization

Translation invariance is achieved using a relative action formulation (Chi et al., 2024) and state-action canonicalization (Zeng et al., 2022; Wang et al., 2021; Zhu et al., 2022a; Jia et al., 2023). We summarize and proof this property.

**Proposition C.3.** *The relative state-action formulation is $\mathbb{T}(3)$ (translational) invariant.*

*Proof.* Translating both the state $S$ and the action $A$ by $g \in \mathbb{T}(3)$ we have:

$$
\begin{align*}
gS^{can} &= \big((O + g) - (e_T + g), \\
&\qquad (e_T + g) - (e_T + g), \\
&\qquad e_R, e_{grip}\big) \\
&= (O - e_T, e_T - e_T, e_R, e_{grip}) \\
&= S^{can} \\
gA^{can} &= \big((A_T + g) - (e_T + g), A_R, A_{grip}\big) \\
&= (A_T - e_T, A_R, A_{grip}) \\
&= A^{can}
\end{align*}
$$

Therefore $\pi(S^{can}) = \pi(gS^{can}) = gA^{can} = A^{can}$. $\qquad \square$

## D. Hyperparameters

Hyperparameters for diffusion-based baseline methods are listed in Table A2. SDP generally adopts Diffusion Policy's hyperparameters, except for batch size, because SDP is heavier than other baselines.

|  | SDP | EquiDiff | EquiBot | DiffPo | DP3 | DP3 paper |
|---|---|---|---|---|---|---|
| Batch Size | 32 | 64 | 64 | 64 | 128 | 128 |
| Prediction Horizon | 16 | 16 | 16 | 16 | 16 | 16 |
| Action Horizon | 8 | 8 | 8 | 8 | 8 | 8 |
| Learning Rate | 1e-4 | 1e-4 | 1e-4 | 1e-4 | 1e-4 | 1e-4 |
| Epochs | 500 | 500 | 500 | 500 | 500 | 3000 |
| Learning Rate Scheduler | cosine | cosine | cosine | cosine | cosine | cosine |
| Noise Scheduler | DDPM | DDPM | DDPM | DDPM | DDIM | DDIM |
| Diffusion Train/Test Step | 100 | 100 | 100 | 100 | 100/10 | 100/10 |
| Encoded Scene Dimension | 128 | 128 | 128 | 128 | 64 | 64 |

*Table A2.* Hyperparameters for baselines.

## E. Architecture of the Point Cloud Encoder

The point cloud encoder is a 5-layer ResNet (He et al., 2016), consisting of EquiformerV2 (Liao et al., 2024) graph convolution layers in the hidden layers and EquiformerV2 origin convolution layer in the last layer to aggregate all the features into a single point.

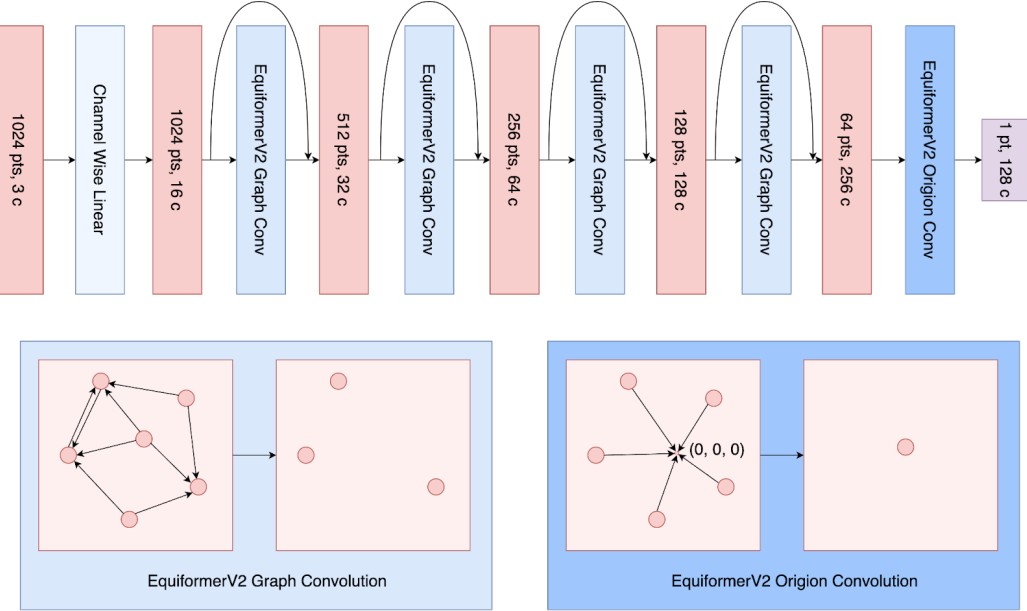

*Figure A5.* Overview of the Point Cloud Encoder. Top: the point cloud encoder. Bottom: the details of each block in the encoder. "pts" stands for the number of points and "c" stands for the number of channels.

