# OpenReview forum: "SE(3)-Equivariant Diffusion Policy in Spherical Fourier Space"
_ICML.cc/2025/Conference — ICML 2025 poster_

### Official Review · Reviewer_Kgae · 2025-02-17

**Overall Recommendation:** 5

**Summary:**

This paper introduces the Spherical Diffusion Policy (SDP), a novel method for robotic manipulation that enforces continuous SE(3) equivariance with spherical Fourier representations. The authors propose a spherical modification of the FiLM layer commonly used in diffusion-based policy networks, prove the invariance property of their proposed approach, and empirically verified their approach in both real and simulated domains.

## update after rebuttal

Accepted the original version of the paper. No updates to the review are necessary.

**Claims And Evidence:**

The paper claims that enforcing SE(3) equivariance using spherical Fourier features leads to better generalization across varied 3D arrangements and improved sample efficiency. These claims are supported by detailed simulation results.

**Essential References Not Discussed:**

N/A

**Experimental Designs Or Analyses:**

They additionally perform an analysis of how all baselines degrade with increasing initialization noise, showing that SDP degrades the most gracefully. However, it's not 100% clear how these experiments were performed from the description: "We train all the baselines on progressively tilted environments with 100 demonstrations." Does tilt=30 degrees imply that demonstrations are on environments sampled between 0 and 30 degrees? Some clarification on this experiment would be helpful.

**Methods And Evaluation Criteria:**

The authors evaluate their approach on a number of MimicGen tasks and real-world robotics tasks, finding that their method (SDP) outperforms other equivariant and non-equivariant baselines. While it seems the MimicGen environments only change the yaw dimension of the objects, the real world experiments use randomizations along other dimensions as well.

Overall, I believe the set of tasks is sufficient for evaluation. Still, It would be nice to see how this method fares in more cluttered scenes.

**Other Comments Or Suggestions:**

Grammatical Issues:

“… the proof essentially follows Schur’s lemma (Schur, 1905) that any linear operation of SO(3) irreps acts as on each irreducible subspace is equivariant.”

“Diffser” → “Diffuser”

“Diffusor” → “Diffuser”

**Other Strengths And Weaknesses:**

Strengths:
- Novel algorithm for SE(3) invariant closed loop control for robot manipulation
- Extensive evaluation and demonstration of the benefits of this approach
- Detailed theoretical justification for the approach

Weaknesses:
- The paper is a bit densely written at several points, making it not very approachable
- No testing in cluttered scenes
- No statistical analysis of the results or inference timing data reported

**Questions For Authors:**

EquiBot (Yang et al., 2024a) are limited to degree l = 1 representation that suppress rich information. Can the authors elaborate on this point and give some examples of problems for which that would be a limiting constraint?

The authors claim that their work is more “computationally efficient” than competing approaches, but fail to provide any inference-time comparisons.

It was not clear to me why equation 5 was chosen in particular. It’s clear from the proof in A.2 that it preserves equivariance, but it seems other expressions (e.g. without normalization) would achieve the same goal.

**Relation To Broader Scientific Literature:**

This work extends and improves upon recent advances in diffusion-based policy learning and equivariant neural networks. It builds directly on prior works such as Diffusion Policy (Chi et al., 2023), EquiDiff, and EquiBot, addressing limitations related to discretized equivariance and computational inefficiencies.

**Theoretical Claims:**

The paper makes claims about the equivariance of the convolution and SFiLM, which I verified in the appendix.

---

> ### Author Rebuttal · Authors · 2025-04-01
>
> We thank the reviewer for their thoughtful feedback. We respond below:
>
> > "However, it's not 100% clear how these experiments were performed from the description: "We train all the baselines on progressively tilted environments with 100 demonstrations." Does tilt=30 degrees imply that demonstrations are on environments sampled between 0 and 30 degrees? Some clarification on this experiment would be helpful."
>
> We are sorry for the confusion. Tilt=30 degrees imply the environment (the table top) is sampled between 0 and 30 degrees. We will add the clarification to the paper.
>
> > "The paper is a bit densely written at several points, making it not very approachable"
>
> Could the reviewer please point out which points can be improved?
>
> > "No testing in cluttered scenes"
>
> For clarification, the (i) Pick Place task and (j) Kitchen tasks in MimicGen (see Figure R3 for visualization) involve multiple objects and can be viewed as slightly cluttered scenes. Unfortunately we don’t have time to test densely cluttered scenes.
>
> > "No statistical analysis of the results or inference timing data reported"
> > "The authors claim that their work is more “computationally efficient” than competing approaches, but fail to provide any inference-time comparisons."
>
> The step-wise success rate for physical experiments is reported in Table A.1. Moreover, in Table R1 we add the inference timing data.
>
> > "Grammatical Issues"
>
> Thank you for pointing out the grammar issues and typos, we will fix these errors in the revision.
>
> > "EquiBot (Yang et al., 2024a) are limited to degree l = 1 representation that suppress rich information. Can the authors elaborate on this point and give some examples of problems for which that would be a limiting constraint?"
>
> Please see the [response](https://openreview.net/forum?id=U5nRMOs8Ed&noteId=rKiX1W2hTw) to ""Although demonstrated in Sec. 4.2, the practical advantages of the proposed ... remain unclear. Specifically, what does it mean that "Vector Neuron only supports up to l=1"?""
>
> > "It was not clear to me why equation 5 was chosen in particular. It’s clear from the proof in A.2 that it preserves equivariance, but it seems other expressions (e.g. without normalization) would achieve the same goal."
>
> We agree other expressions (without normalization or using tensor products, etc) could achieve the same equivariance. The proposed method stems from the nonlinearity operation in Vector Neurons. We find Equation 5 works well and we haven’t compared other expressions.

---

### Official Review · Reviewer_Sqpd · 2025-03-10

**Overall Recommendation:** 2

**Summary:**

This paper propose one new method called “Spherical Diffusion Policy (SDP)” for robot manipulation. The paper focus on 3D generalization, using SO(3) and T(3) equivariance (i.e., full SE(3) group) to handle random tilts and object placements. It design a special spherical encoder with spherical FiLM layer and “spherical denoising temporal U-net (SDTU).” The method is tested on many simulation tasks as well as real robot tasks (both single-arm and bimanual). The result show improved performance compared to baselines like EquiDiff or Diffusion Policy in environment with random initial poses.

## After Rebuttal
Though the overall results reported by authors are very good, in the authors' rebuttal, they are not able to reply my concern that the baseline is not tuned. The authors provide the param table while they deliberately miss the baseline (DP3) that is not tuned in the table. Also the reply from authors "We use the DP3 results reported in the CoRL paper Equivariant Diffusion Policy" show that the results are directly copied from previous works without justification. I am not sure if there is an issue on this but this looks suspicious to me. Considering the above mentioned factors, I would change my score from weak accept to weak reject. The other parts of this paper are all good.

**Claims And Evidence:**

The claims are supported by simulation and real-world experiments.

**Essential References Not Discussed:**

No

**Experimental Designs Or Analyses:**

The experiment analysis overall looks good to me, with an extensive baseline comparison and a few ablation studies.
However, there is a small problem regarding to baselines. I found that the results for DP3 (Ze et al., 2024) do not make sense to me, as it uses 3D information yet even worse than image-based methods, which does not align with my tuning experience. Could authors tune some parameters of this baseline to see, like the longer prediction horizon? Besides, it is good to report the parameters for all baselines.

**Methods And Evaluation Criteria:**

The proposed method and evaluation criteria make sense to me.

**Other Comments Or Suggestions:**

1. More ablation studies would be helpful to understand the method.
2. Detailed parameters for baselines for fair comparison.

**Other Strengths And Weaknesses:**

N/A

**Questions For Authors:**

As mentioned above, the authors are suggested to tune the baseline methods or report the failure mode of the baselines/their own method. Besides, how the baseline is implemented and tuned should be reported.

**Relation To Broader Scientific Literature:**

The contributions are very related to the literature.

**Theoretical Claims:**

Yes.

---

> ### Author Rebuttal · Authors · 2025-04-01
>
> We thank the reviewer for their thoughtful feedback. We respond below:
>
> > "Could authors tune some parameters of this baseline to see, like the longer prediction horizon? Besides, it is good to report the parameters for all baselines."
>
> We use the DP3 results reported in the CoRL paper Equivariant Diffusion Policy. Additionally, we implemented a new baseline, DP3-canonicalization, which improves upon DP3 but still significantly underperforms SDP (see Table R2). Regarding DP3's poor performance on the MimicGen tasks, we hypothesize that this is due to its use of an MLP as the vision encoder. The MLP may struggle to capture information about multiple objects or their orientations—factors that are critical in the MimicGen tasks.
>
> > "More ablation studies would be helpful to understand the method."
>
> In Table R2, we add DP3 (no equivariance) and DP3-cano (global SE(3)-equivariance) as additional baselines for ablation. We find that SDP, which has local equivariance, outperforms the global SE(3)-equivariance used in DP3-cano.
> We also introduce a new baseline, Discrete SDTU, which enforces discrete equivariance using the Octahedral or Cubic group, which discretize SO(3)—similar to EquiDiff, which uses the cyclic group C8 that discretize SO(2). Our results show that Discrete SDTU leads to an approximate 10% drop in performance, highlighting the advantage of SDP’s use of continuous spherical Fourier representations.
>
> > "Detailed parameters for baselines for fair comparison."
>
> We provide detailed parameters for baselines. SDP generally adopts Diffusion Policy’s hyperparameters, except for batch size, because SDP is heavier.
>
> Table R3. Hyperparameters for baselines.
> |                           | SDP    | EquiDiff | EquiBot | DP     |
> |---------------------------|--------|----------|---------|--------|
> | Batch Size                | 32     |    64    |    64   |   64   |
> | Prediction Horizon        |   16   |    16    |    16   |   16   |
> | Action Horizon            |    8   |     8    |    8    |    8   |
> | Learning Rate             |  1e-4  |   1e-4   |   1e-4  |  1e-4  |
> | Epochs                    |   500  |    500   |   500   |   500  |
> | Learning Rate Scheduler   | cosine |  cosine  |  cosine | cosine |
> | Noise Scheduler           |  DDPM  |   DDPM   |   DDPM  |  DDPM  |
> | Diffusion Train/Test Step |   100  |    100   |   100   |   100  |
> | Encoded Scene Dimension   |   128  |    128   |   128   |   128  |
>
> > "report the failure mode of the baselines/their own method."
>
> Detailed failure modes are reported in Section B.3, Table A1, and Figure A4. The major failure mode is inaccurate action prediction when the end effector is about to make contact. We will strengthen the connection between the failure modes analysis section and the main text.

---

### Official Review · Reviewer_CbPJ · 2025-03-11

**Overall Recommendation:** 3

**Summary:**

This paper works on improving the equivariance of diffusion policy for manipulation tasks. To this end, the authors align the input point clouds and robot state into a canonical coordinate frame to achieve translational equivariance, and then project the encoded observations onto spherical harmonic basis to achieve SO(3) equivariance. To further handle the SO(3) equivariance within network structure itself, the authors upgrade the 1D convolution in vanilla diffusion policy to a mix channel temporal convolution, and the FiLM condition to a spherical FiLM layer. To demonstrate the effectiveness of the proposed method, especially its SE(3) equivariance, the authors augment the scene to manipulate with additional rotation and translation,  and achieved superior performance than existing baselines that use absolute control or velocity control mode.

**Claims And Evidence:**

The major claim of this paper is its SE(3) equivariance and its better sample efficiency and generalization. These claims are in general well supported with straight forward proof of propositions as well as extensive experiments.
However, I have two major concerns regarding to this claim:
- First, do we really have to achieve SO(3) equivariance within the network architecture itself when the observation inputs are point clouds?
  - If the author target to achieve SO(3) equivariance to relative rotation to the end-effector, the point clouds and robot arm states can be further aligned within gripper's local coordinate frame by applying a known rotation.
  - If the authors target to achieve SO(3) equivariance to object's absolute rotation, then it is equivariant when the scene is under a global rotation to a demonstration. However, during the experiments, the proposed method demonstrate significantly better performance on manipulation tasks with multiple rigid objects, which needs more justification.
- Second, the translation equivariance is achieved by explicit alignment of point clouds and proprioceptive states, therefore the diffusion policy network itself is SO(3) equivariant. Normalization is a standard preprocessing in point cloud networks, therefore it is better to explain how this normalization is distinct from others. In general, I prefer to have SO(3) in the title instead of SE(3) to be more focused and reduce confusion.

This paper needs more discussions and justifications about these two issues that will better validate the motivation of this paper.

**Essential References Not Discussed:**

Essential references have been discussed to understand this paper.

**Experimental Designs Or Analyses:**

The experiment designs are valid except for two minor issues:
- First, the paper only compare with previous works with 100 demonstrations for training, it worth to conduct experiments on more demonstrations, such as the settings in EquiDiff to see how the performance saturate.
- Second, in Sec.5.3 it is somewhat surprising that absolute position control is significantly worse. However, there is no further elaboration about the detailed settings, such as wether the control signal is directly from the policy network or converted from relative control signals. In addition, it is also unclear what are the conclusion we can draw from this performance gap.

**Methods And Evaluation Criteria:**

The methods and evaluation criteria make sense in general, except for one minor issue:
- Projection on to spherical harmonic basis will lose details of added noise and therefore potentially interferes the diffusion process. Therefore, it would be beneficial to discuss it and correspondingly validate the maximum degree of spherical harmonics in ablation studies.

**Other Comments Or Suggestions:**

- Is that better to remove e_i^T−e_i^T since it is constantly zero, or is there any specific reason to keep it?

**Other Strengths And Weaknesses:**

Strengths:
 - The performance is significantly better than existing work when number of demonstration is limited for training.

Weaknessnes:
- Some descriptions needs to be further elaborated: line-236~240: The architecture of the encoder is difficult to understand. Does the authors mean that a resnet is applied point-wisely first and then sent to EquiformerV2 for further feature extraction? It is better to illustrate the detail of encoder as well to reduce confusion.
- Any motivation to have physical experiments with only one ray-fin finger as the end-effector on each arm? Which is less consistent to the simulation.

**Questions For Authors:**

Please refer to the above comments and address major concerns about:
- Claims And Evidence: what are the specific rotation this paper aims to be equivariant to, transformation between gripper coordinate frame, global transformation of the scene, or individual transformation of objects within a scene
- Experiments: Discuss how relative position control and SE(3) equivariance contribute to the performance distinctly and their relation. Also discuss about how the performance varies when more demonstration is given.
- Methods: Elaborate more about the structure of the encoder and the projection onto spherical harmonics's effect on the diffusion process.

**Relation To Broader Scientific Literature:**

This paper is an application of SO(3) equivariance to diffusion policy tasks. Therefore, it has connection to previous works about equivariant networks. While the results demonstrated in this paper is domain-specific and does not expand the understanding of equivariant network architectures in general.

**Theoretical Claims:**

I checked the propositions in this paper and they are generally correct.
My only concern is that theoretically the policy is only SO(3) equivariant to scene under a global rotation. However, the experiments demonstrate superior performance in scene with multiple randomly placed objects, such as Three Pc. Assembly, which requires more justification and explanation.

---

> ### Author Rebuttal · Authors · 2025-04-01
>
> We thank the reviewer for their thoughtful feedback. We respond below ([link to figures](https://limewire.com/d/DAndu#qMh5UOCXI6)):
>
> >"... the point clouds and robot arm states can be further aligned within gripper's local coordinate frame by applying a known rotation.
> If the authors target to achieve SO(3) equivariance to object's absolute rotation, then it is equivariant when the scene is under a global rotation to a demonstration."
>
> Leveraging SE(3)-invariant representations is a valuable baseline suggestion, and we have included it as an ablation. Specifically, we implemented a baseline called DP3-cano, which achieves SE(3)-invariance by canonicalizing (i.e., normalizing or transforming) both the point cloud and the action into the gripper frame. However, experimental results show that DP3-cano significantly underperforms the equivariant method SDP.
>
> Table R2. Additional Ablations.
> |                 |                         | Coffee 15 | Three Pc. As. 15 | Square 15 | Threading 15 | Avg SR |
> |-----------------|-------------------------|-----------|------------------|--------|-----------|--------|
> | SDP             | SE(3) equivariance      | 54        | 49               | 38     | 53        | 49     |
> | Discrete U-net* | Octahedron equivariance | 42        | 16               | 34     | 48        | 35     |
> | DP3-cano*       | SE(3) equivariance        | 40        | 0                | 8      | 12        | 15     |
> | DP3*            | None                    | 20        | 0                | 0      | 4         | 6      |
>
> *Currently, these baselines have been trained for 300 out of the planned 500 epochs. We expect the final average success rate to improve by approximately 4%.
>
> We hypothesize that local equivariance in SDP plays a crucial role. Furthermore, Benjamin et al. [1] demonstrate that using equivariant features results in significantly lower prediction errors compared to invariant features in molecular property prediction tasks.
>
> [1] Benjamin Kurt Miller, Mario Geiger, Tess E. Smidt, Frank Noé, Relevance of Rotationally Equivariant Convolutions for Predicting Molecular Properties, Machine Learning for Molecules Workshop at NeurIPS 2020
>
> > "Second, the translation equivariance is achieved by explicit alignment of point clouds ... Normalization is a standard preprocessing in point cloud networks, ... I prefer to have SO(3) in the title instead of SE(3) to be more focused and reduce confusion."
>
> For clarification, our method is SE(3)-equivariant. Specifically, SDP achieves SO(3) equivariance through the use of spherical Fourier representations, and T(3) (translation) equivariance via canonicalization. It is important to note that canonicalization differs from normalization in that the generated action is also transformed into the canonical frame.
>
> > "it would be beneficial to discuss it and correspondingly validate the maximum degree of spherical harmonics in ablation studies."
>
> We agree that truncated spherical harmonic coefficients approximate the underlying spherical function, and using a low maximum frequency can lead to a loss of important details. As shown in Table A3 of the paper, setting $l = 1$ results in a noticeable performance drop. However, performance saturates at $l = 2$ and $l = 3$, suggesting that higher frequencies provide diminishing returns. Due to space limitations, this table is currently located in the appendix, but we will strengthen the connections in the main text to highlight this finding.
>
> > "... it worth to conduct experiments on more demonstrations, such as the settings in EquiDiff to see how the performance saturate."
>
> This is a great question—we have plotted data scaling curves in Figure R1. Each point represents the average performance across four tilted-table tasks (with tilt angles in $[0, 15^\circ]$) for SDP, EquiDiff (EDP), and DiffPo (DP). Notably, SDP trained with $10^2$ demonstrations outperforms EDP trained with $10^3$, while EDP with $10^2$ demonstrations achieves comparable performance to DP trained with approximately $10^{2.5}$ demonstrations.
>
> > "... no further elaboration about the detailed settings, such as wether the control signal is directly from the policy network or converted from relative control signals."
>
> The absolute position control does not have translational equivariance while relative position control does, as explained in Section 4.1 and proof in Appendix C.2.
>
> The ablation study in Section 5.3, which uses absolute action control, highlights the importance of translational equivariance in our method.
>
> > "The architecture of the encoder"
>
> See Figure R2.
>
> > "Any motivation to have physical experiments with only one ray-fin finger ..."
>
> We use the shared robot in our lab and didn’t make any changes on the hardware.
>
> > "Is that better to remove e_i^T−e_i^T"
>
> Yes, it's for the convenience of presentation in the paper.

---

### Official Review · Reviewer_mcma · 2025-03-14

**Overall Recommendation:** 3

**Summary:**

This paper presents a novel SE(3)-equivariant diffusion policy, named Spherical Diffusion Policy (SDP), aimed at improving generalization for robotic manipulation tasks across varying 3D transformations. The key motivation stems from the assumption that embedding states, actions, and denoising processes in spherical Fourier space ensures SE(3)-equivariance, thereby enabling robust generalization across transformed scenes without extensive data collection. Specifically, the framework incorporates a spherical encoder to embed scene features, spherical FiLM layers for equivariant conditioning, and a spherical denoising temporal U-net for spatiotemporal equivariance. Additionally, theoretical analyses verify the equivariance of the proposed method. Extensive simulation and physical experiments demonstrate the effectiveness and superior performance of SDP over state-of-the-art baselines on multiple challenging robot manipulation tasks.

**update after rebuttal**

After reading the rebuttal, most of my concerns have been addressed, and I am inclined to keep my original score.

**Claims And Evidence:**

Yes.

**Essential References Not Discussed:**

Some SE(3) Diffusion Models should be cited as the related work, such as:
[1] SE (3) diffusion model with application to protein backbone generation, ICML'2023
[2] SE (3) diffusion model-based point cloud registration for robust 6d object pose estimation, NeurIPS'2023
[3] SE(3)-DiffusionFields: Learning smooth cost functions for joint grasp and motion optimization through diffusion, ICRA'2023

**Experimental Designs Or Analyses:**

Yes.

**Methods And Evaluation Criteria:**

Yes.

**Other Comments Or Suggestions:**

See weaknesses.

**Other Strengths And Weaknesses:**

1. Strengths
- The paper introduces an innovative SE(3)-equivariant diffusion policy framework, effectively enabling model generalization across SE(3)-transformed environments;

- The proposed spherical Fourier features, spherical FiLM layers, and spherical denoising temporal U-Net achieve SE(3)-equivariance with solid theoretical grounding;

- Extensive empirical evaluations in both simulation and real-world robotic experiments demonstrate substantial improvements in performance and generalization compared to existing state-of-the-art methods.

2. Weaknesses
Although demonstrated in Sec. 4.2, the practical advantages of the proposed spherical Fourier representations over previous SE(3)-equivariant representations (e.g., Vector Neuron and ET-SEED) remain unclear. Specifically, what does it mean that "Vector Neuron only supports up to l=1"? Furthermore, could you clarify how the "truncated spherical Fourier coefficients provide a compact approximation of spherical features and are compatible with SO(3) rotations," in contrast to the "computationally heavy SO(3) irreps used in ET-SEED"?

- Table 1 indicates that as the degree of SE(3) initialization increases, SDP continues to exhibit significant performance degradation, raising concerns about the actual effectiveness and robustness of the proposed method. Could you address this point explicitly?

- Please clarify the fundamental differences between the spherical Fourier representations employed in this paper and those used in Spherical Fourier Neural Operators.

- Given the importance of inference speed in robotic control applications, comparisons regarding computational efficiency are currently missing. It would be beneficial if the authors could provide clarity on this aspect.

**Questions For Authors:**

Why not directly consider SE(3)-invariant representations rather than focusing on designing SE(3)-equivariant representations? Both approaches yield similar results, and SE(3)-invariant representations might even be better designed than equivariant ones？

**Relation To Broader Scientific Literature:**

The paper advances prior works on equivariant diffusion policies by generalizing leveraging spherical Fourier representations for improved generalization in robotic manipulation tasks.

**Theoretical Claims:**

Yes.

---

> ### Author Rebuttal · Authors · 2025-04-01
>
> We thank the reviewer for their thoughtful feedback. We respond below:
>
> > "Some SE(3) Diffusion Models should be cited as the related work, such as: ..."
>
> Thank you for highlighting these relevant works. We will include citations to these papers in the revised related work section.
>
> >"Although demonstrated in Sec. 4.2, the practical advantages of the proposed ... remain unclear. Specifically, what does it mean that "Vector Neuron only supports up to l=1"?"
>
> Both Vector Neurons (VN) and our method can be interpreted as representing features in spherical Fourier space, up to a specified maximum frequency $l$. While our method supports arbitrary spherical harmonic types (we use $l_{max}=2$ in our paper), VN is limited to using only scalars ($l=0$) and vectors ($l=1$) as features. The scalars in VN are mathematically equivalent to type-0 features, which are invariant to rotation. The 3D vectors $V = [V_x, V_y, V_z]$ used in VN correspond to type-1 features $c_1 = [c^{-1}_1, c^{0}_1, c^{1}_1]$, as both are three-dimensional and transform under rotation via a rotation matrix. Thus, VN only supports spherical features up to type-1. However, type-1 features have limited representational capacity—for example, they are incapable of capturing spherical distributions with two distinct modes.
>
> >" could you clarify how the "truncated spherical Fourier coefficients ..." in contrast to the "computationally heavy SO(3) irreps used in ET-SEED"?"
>
> Both SDP and ET-SEED achieve SE(3) equivariance by assigning SO(3)-steerable features to each point in the point cloud. While both methods use irreducible representations (irreps), SDP employs lightweight, order-wise linear layers—each order $m$ only connects to itself. In contrast, ET-SEED (based on the SE(3)-Transformer) utilizes heavier and more redundant fully connected linear layers that connect all orders $m$ across all types $l$.
>
> > "...comparisons regarding computational efficiency are currently missing."
>
> Empirically, although ET-SEED employs a two-stage diffusion process, its inference time is 60× slower than that of SDP (29.4s vs. 0.44s; see Table below). Moreover, SDP's inference time is on the same order of magnitude as that of Diffusion Policy (SDP is approximately 5× slower than DP).
>
> Table R1. Practical statistics for SDP and SOTA baselines.
> |                            | Diffusion Policy | EquiDiff                 | EquiBot                    | SDP                        | ET-SEED                           |
> |----------------------------|------------------|--------------------------|----------------------------|----------------------------|-----------------------------------|
> | Inference Speed (Second) ↓ | 0.09             | 0.14                     | 0.18                       | 0.44                       | 29.4                              |
> | Training Batch Size ↑      | 64               | 64                       | 64                         | 32                         | 1                                 |
>
>
> > "Table 1 indicates that as the degree of SE(3) initialization increases, SDP continues to exhibit significant performance degradation, .... Could you address this point explicitly?"
>
> Authors agree that SE(3) equivariant methods should perform consistently no matter how the task is rotated or translated in 3D, provided the transformations are of the full environment. However, in Table 1, these tasks are not perfectly SE(3) transformed, since the gravity direction is not transformed, the camera view (occlusion) is not transformed, and the robot (kinematics) is not transformed. All of these factors could add complexity to the task (e.g., greater table tilt leads to increased object instability), thus SDP continues to exhibit performance degradation. Despite the increasing complexity, we find that SDP outperforms all baselines on all levels of SE(3) initialization of the tilted table tasks in Table 1.
>
> > "Please clarify the fundamental differences ... in Spherical Fourier Neural Operators."
>
> Both methods leverage the spherical Fourier (SF) transform to process signals on the sphere. However, our method is purely based on SF, while SFNO combines an SF neural network for global convolution with a pointwise MLP for local non-linearity. Additionally, our proposed SDTU enables SO(3) and temporal equivariant convolution, and SFiLM provides equivariant spherical conditioning—capabilities not present in SFNO.
>
> > "Why not directly consider SE(3)-invariant representations?"
>
> Leveraging SE(3)-invariant representations is a valuable baseline suggestion, and we have included it as an ablation. Specifically, we implemented a baseline called DP3-cano in [Table R2](https://openreview.net/forum?id=U5nRMOs8Ed&noteId=IN1AESox8P), which achieves SE(3)-invariance by canonicalizing (i.e., normalizing or transforming) both the point cloud and the action into the gripper frame. However, experimental results show that DP3-cano significantly underperforms the equivariant method SDP.

---

### Decision · Program_Chairs · 2025-05-01

**Decision:**

Accept (poster)

**Comment:**

This paper proposes a novel method, Spherical Diffusion Policy, which is equivariant to 3D rotations and invariant to 3D translations enabling generalization to unseen scenes. Specifically, it combines a spherical encoder to embed scene features, spherical FiLM layers for equivariant conditioning, and a spherical denoising temporal U-net for spatiotemporal equivariance. The equivariance of the proposed method is verified both theoretically and empirically. While most of the reviewers appreciate the novelty of the paper, one reviewer raised concern on the parameters of one baseline. However, overall the experiments are sound, and I recommend an accept.